# DaDA: Distortion-aware Domain Adaptation for Unsupervised Semantic Segmentation

**Sujin Jang**
Samsung Advanced Institute of Technology
s.steve.jang@samsung.com

**Joohan Na**
Samsung Advanced Institute of Technology
joohan.na@samsung.com

**Dokwan Oh**
Samsung Advanced Institute of Technology
dokwan.oh@samsung.com

## Abstract

Distributional shifts in photometry and texture have been extensively studied for unsupervised domain adaptation, but their counterparts in optical distortion have been largely neglected. In this work, we tackle the task of unsupervised domain adaptation for semantic image segmentation where unknown optical distortion exists between source and target images. To this end, we propose a **d**istortion-**a**ware **d**omain **a**daptation (DaDA) framework that boosts the unsupervised segmentation performance. We first present a **r**elative **d**istortion **l**earning (RDL) approach that is capable of modeling domain shifts in fine-grained geometric deformation based on diffeomorphic transformation. Then, we demonstrate that applying additional global affine transformations to the diffeomorphically transformed source images can further improve the segmentation adaptation. Besides, we find that our distortion-aware adaptation method helps to enhance self-supervised learning by providing higher-quality initial models and pseudo labels. To evaluate, we propose new distortion adaptation benchmarks, where rectilinear source images and fisheye target images are used for unsupervised domain adaptation. Extensive experimental results highlight the effectiveness of our approach over state-of-the-art methods under unknown relative distortion across domains. Datasets and more information are available at https://sait-fdd.github.io/.

## 1 Introduction

Recent years have witnessed dramatic improvements in semantic segmentation performance based on a massive amount of pixel-level annotations. However, applying a segmentation model learned from the training dataset (source domain) directly to unseen test scenarios (target domain) often leads to significant performance degradation, which is caused by the divergence of data distribution between source and target domains. In addition, manually annotating target domain images with pixel-wise labels is extremely expensive and taxing. To address such issues, many studies have proposed unsupervised domain adaptation methods to transfer the knowledge learned from the labeled source domain to the unlabeled target domain (*e.g.,* [33, 34, 27, 26, 40]).

Existing works on unsupervised segmentation adaptation methods have predominantly investigated domain shift problems mainly due to photometric and textural discrepancies between rectilinear image domains (*e.g.,* adapting synthetically generated images [29, 30] to real-world imagery [8]; or adapting images from/to different cities [7]). In contrast, domain shifts in geometric and optical distortion have not been well explored, despite commonly appearing in many practical applications. For example, wide-angle cameras (*e.g.,* fisheye cameras) have been extensively used for complex

36th Conference on Neural Information Processing Systems (NeurIPS 2022).

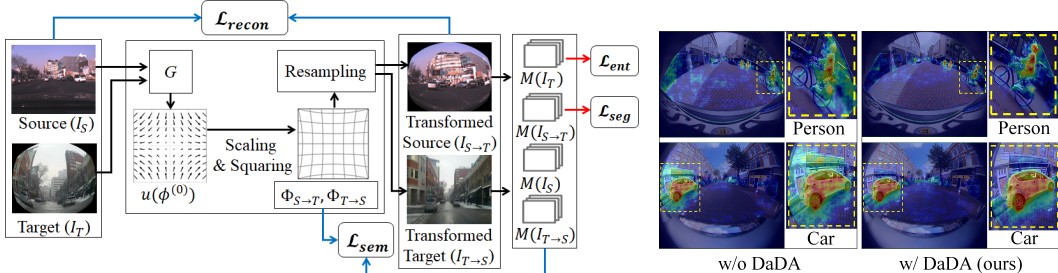

(a) Distortion-aware domain adaptation (DaDA) framework.     (b) Class-activation visualizations [31].

Figure 1: **Overview of DaDA framework and qualitative results. (a)** We depict the relative distortion learning (blue) and the segmentation adaptation (red) with related loss functions. Deformation field generator $G$ produces a flow field $u(\phi^{(0)})$ to transform source data ($I_S$) to a new image ($I_{S \to T}$) replicating the distortion style of the target image ($I_T$). **(b)** Our framework shows stronger and finer boundary of class-wise activation visualizations under unknown radial distortion across domains (more illustrations are shown in Appendix C).

vision systems like autonomous vehicles [38] and surveillance [4] to obtain more information about the surrounding environment. Fisheye images (*e.g.,* target image $I_T$ in Fig.1) have quite different geometric deformations (*e.g.,* radial distortion) compared to regular rectilinear images (*e.g.,* source image $I_S$ in Fig.1). Such distortion variants pose even more challenging domain adaptation tasks. For example, a state-of-the-art semantic segmentation model [6] only trained on rectilinear images fails to correctly predict pixels of vehicles and roads under optical distortion and its overall prediction accuracy drops from $67.02\%$ (trained on fisheye images) to $32.39\%$ (trained only on rectilinear images, see Tab.1). To alleviate this, we may rectify distortion at test time. However, such a workaround inevitably leads to reduced field-of-view (*e.g.,* over $30\%$ loss for fisheye images [38]), resampling distortion artifacts at the image periphery, calibration errors in practice, and additional computing resources for the rectification step at test time [19], which is against the original purpose of using wide-angle cameras. This urges us to use native fisheye images without needing rectification. Another motivation for our work is the scarcity and difficulty of constructing annotations for distorted fisheye images (*e.g.,* Woodscape [38]), while we already have larger amounts of annotations for rectilinear images (*e.g.,* Cityscapes [8], GTAV [29]). Remarkably, we have relatively fewer public datasets providing large-scale and finely annotated fisheye images. Woodscape [38] is currently the only real-world public dataset with segmentation annotations. Such a lack of annotations for distorted images necessitates introducing optical distortion into unsupervised domain adaptation.

With these insights, we first formulate an important but challenging unsupervised domain adaptation task for semantic segmentation where unknown optical distortion exists between the source and target domains. To this end, we propose a novel *distortion-aware domain adaptation* (DaDA) framework, which provides a new perspective to minimizing domain gaps in geometric and optical distortion. We first present *relative distortion learning* (RDL), which is capable of modeling relative deformation between the source and target domain. In particular, we build a deformation field generator to transform the source image to a new image sharing similar distortion features of the target image. To enable such challenging unsupervised and unpaired distortion learning, we exploit the properties of *diffeomorphism*, that is differentiable and has a differentiable inverse. We directly integrate such properties into our *distortion-aware losses* to enforce the semantic quality of relative deformation fields at the image- and the prediction level. We also observe that applying additional global affine transformations (*e.g.,* rotation, shearing) to the diffeomorphically transformed source images can further improve the segmentation adaptation performance in most cases. Owing to fine-grained diffeomorphic and global affine transformations, our framework provides distortion-aware segmentation adaptation models and reliable pseudo labels, and thus ultimately improves the performance of self-supervised learning methods.

To validate, we propose new domain adaptation benchmarks, where a segmentation model trained on rectilinear images (Cityscapes [8] or GTAV [29]) is transferred to fisheye images (Woodscape [38] or our in-house fisheye driving scene dataset (FDD)). With mean Intersection-over-Union (mIoU) as the evaluation metric, our framework achieves significantly improved prediction performance compared to existing segmentation adaptation methods.

In essence, our key contributions can be summarized as: (1) a distortion-aware domain adaptation framework for boosting unsupervised semantic segmentation performance in the presence of unknown relative distortion; (2) distortion-aware losses that effectively enforce the unpaired and unsupervised relative distortion learning to transfer distortion style from target to source domain; (3) new unsupervised domain adaptation benchmarks posing challenging tasks, where source and target images have additional domain gaps in optical distortion.

## 2 Related Work

### 2.1 Domain Adaptive Semantic Segmentation

Recent advances in image-to-image translation and style transfer [42] have promoted the translation of source-domain images to mimic texture and appearance of the target-domain images. CyCADA [24] applied CycleGAN [42] to preserve semantic consistency before and after image-level adaptation. BDL [22] proposed a bidirectional learning method where image-to-image translation and segmentation adaptation is learned alternatively. Similar methods have been proposed to adapt source images [35, 17] or to reconstruct the source image from the output-level representation [36]. Deviated from these adversarial approaches, FDA [37] proposed a method to align the low-level frequencies of source images and their counterparts in target images. Ma et al. [25] applied a photometric alignment method to minimize domain shifts in image- and category-level feature distributions.

More recently, self-supervised learning (SSL) approach has emerged in domain adaption tasks to further improve the segmentation adaptation performance. SSL approaches try to rectify noisy pseudo labels with respect to uncertainties or confidence scores [22, 17, 28, 27, 26, 13, 41, 40]. These methods inherently require good initial models, that are commonly trained on source or adapted on target beforehand. To obviate the need of multi-stage and adversarial training, Araslanov and Roth [2] introduced a photometric augmentation method to SSL.

However, none of the prior works directly considers distortion-oriented domain shifts in the unsupervised segmentation adaptation tasks. Moreover, it remains unknown how well existing adaptation methods address unknown relative distortion between domains. As a pioneering work, we first aim to evaluate prior arts on the adaption task in the presence of unknown optical distortion across domains. For this, we noted that many of current state-of-the-art methods rely on SSL (*e.g.,* [17, 27, 26, 41, 40]) and have adopted AdaptSeg [33] or AdvEnt [34] as for their based segmentation adaptation. Thus, we take these adversarial adaptation approaches as our baseline methods and further evaluate SSL approaches [26, 27, 40] in the presence of optical distortion shifts across domains. To evaluate, we newly formulate segmentation adaptation benchmarks, *i.e.,* transferring from real or synthetic rectilinear images to real fisheye images.

### 2.2 Diffeomorphic Deformation Networks

Spatial transformation networks (STN [15]) have been used in various contexts since it is capable of learning differentiable deformation field. STN primarily allows simple linear transformations (*e.g.,* affine, translations, scaling, and rotations) and can be extended to more flexible mappings such as thin plate spline transformations [23]. However, optical distortion involves complex nonlinear transformation and STNs are limited to support such transformations. Instead Detlefsen et al. [12] proposed to exploit *diffeomorphisms* in spatial transformation, which can model complex nonlinear deformation and address optimization divergence issues in learning spatial transformation. A *diffeomorphism* is a globally one-to-one continuous and smooth mapping, which has a differentiable inverse. Diffeomorphic transformations have been largely used for image registration [16, 10, 32] and shape analysis [39]. However, traditional diffeomorphic deformation methods demand high computational costs and difficult to implement; and thus make it difficult to be incorporated into deep neural networks. To address such issue, Dalca et al. [11] proposed a probabilistic generative model to generate diffeomorphic deformation field for medical image registration task. However, their work depends on a *paired* set of 3D brain images depicting the same contexts. Inspired by [11], we propose a new relative distortion generator, which takes a set of *unpaired* source- and target-domain image, to transform the source image to a new image sharing similar distortion style of the target image.

## 2.3 Radial Distortion Rectification

Traditional distortion rectification approaches require images captured with specific camera calibration parameters [20] and thus are not flexible. The other work exploits the principle that a straight line should be projected into a straight line in a calibrated image [1], but such method mainly depends on the accuracy of line detection. More recently, deep neural networks have been used for robust and efficient rectification performance [5, 21]. However, these works primarily aim to rectify distorted images and do not support distortion style transfer for unpaired and unsupervised domain adaptation tasks. In contrast, we propose to transfer distortion style between unpaired source and target images.

# 3 Method

Let $S$ be the source-domain data with segmentation labels $Y_S$ and $T$ be the target-domain data with no labels. We aim to train a scene segmentation network $M$ performing satisfactory pixel-wise prediction on $T$ by minimizing domain gaps between $S$ and $T$. Here we formulate more challenging adaptation tasks where distributional shifts include not only visual domain gaps (*e.g.,* texture, lighting, contrast) but also geometric and optical distortions (*e.g.,* radial distortion). Such domain gaps make $M$ difficult to learn transferable knowledge in both visual and geometric domains.

## 3.1 Relative Distortion Learning

To minimize the discrepancy between source and target domain, prior works proposed pixel-level image translation methods based on cycle-consistency [14, 17, 35, 42] or photometric alignment [2, 25]. However, these methods do not consider domain gaps in geometric distortion and thus fail to transfer distortion style from $I_T$ to $I_S$ (see Fig.4). To address this, we propose a *relative distortion learning* (RDL) method, which predicts a relative deformation field to transfer the distortion style between domains. Given a source-domain image $I_S$ and a target-domain image $I_T$, we aim to transform $I_S$ to a new image $I_{S \to T}$ based on a relative deformation field $\Phi_{S \to T}$, where $I_{S \to T}$ shares a similar distortion style of $I_T$. We achieve this transformation through a grid-based sampling operation $I_S \circ \Phi_{S \to T} = I_{S \to T}$. Ultimately, the transformed source image $I_{S \to T}$ aim to mitigate the domain shift in optical distortion at a fine-grained level. However, it is not trivial to predict relative deformation fields since input images are not paired (*i.e.,* dissimilar image contents); and the relative distortion involves nonlinear geometric deformation, which is hard to be parameterized without knowing optical features (*e.g.,* lens distortion model, focal length). To learn such a challenging geometric relationship, we exploit *diffeomorphic transformation* in the unpaired and unsupervised distortion translation task.

**Diffeomorphic Transformation.** Let $u \in \mathbb{R}^{2 \times w \times h}$ be a flow field which is constant over time. Then, we describe a differential equation of evolution of deformation $\phi^{(t)}$ by

$$\frac{\partial \phi^{(t)}}{\partial t} = u(\phi^{(t)}), \tag{1}$$

where $t$ is time and $\phi^{(0)}$ the identity transformation. By integrating $u(\phi^{(t)})$ over $t \in [0, 1]$, we get a diffeomorphic deformation field $\Phi := \phi^{(1)}$, which maps the coordinates from one image to another image. For fast and differentiable integration, we use an exponentiated integration technique, so-called *scaling-and-squaring* [3]. This integration method starts from $\phi^{(1/2^T)} = \phi^{(0)} + u/2^T$ and iteratively computes deformation fields of next time steps via $\phi^{(1/2^{t-1})} = \phi^{(1/2^t)} \circ \phi^{(1/2^t)}$ over $T$ times, where $\circ$ indicates a grid-based resampling operation. Then, we obtain an approximate deformation field $\Phi$. The inverse deformation field $\Phi^{-1}$ can be achieved by integrating the negative field $\phi^{(1/2^T)} = \phi^{(0)} - u/2^T$ via the same integration method.

To generate a field of $\Phi_{S \to T}$, which defines relative distortion between $I_S$ and $I_T$, we propose a deformation field generator $G$, which takes both $I_S$ and $I_T$; and their first-order gradient (*i.e.,* Sobel filter) $\nabla I_S$ and $\nabla I_T$ as input. Here we use the image gradients to provide rich geometric features inspired by the distortion rectification method [1]. Then, $G$ generates deformation field $\Phi_{S \to T}$ and its inverse field $\Phi_{T \to S}$. For example, in Fig.2, $G$ generates a flow field $u$ to construct a $\Phi_{S \to T}$ (red-colored grid in (e)) from a pair of $I_T$ (a) and $I_S$ (b). Finally, a new transformed image $I_{S \to T}$ (e) is generated via $I_S \circ \Phi_{S \to T}$. Similarly, negative integration of the flow field $u$ yields an inverse deformation field $\Phi_{T \to S}$, which generate a reconstructed image (d) via $I'_S = I_{S \to T} \circ \Phi_{T \to S}$.

To supervise the generator $G$ for learning the relative distortion in unpaired and unsupervised settings, we exploit the properties of diffeomorphic transformations: *topology preserving, invertible*, and *differentiable*. Such diffeomorphic constraints are directly built into relative distortion learning to enforce desirable deformation field outputs. In particular, we propose distortion-aware losses: *distortion reconstruction* loss and *semantic distortion-consistent* loss. These losses evaluate the cycle-consistency of the relative deformation field at the image- and the prediction-level, while enforcing the semantic quality of the relative deformation field. Distortion reconstruction loss ($\mathcal{L}_{recon}$) ensures that $G$ generates convincing flow fields to reconstruct the source image $I_S$ from the transformed image $I_{S \to T}$ by the inverse deformation field $\Phi_{T \to S}$, and vice versa:

$$\mathcal{L}_{recon} = \|I_S - I'_S\|_1 + \|I_T - I'_T\|_1, \text{ where } I'_S = I_{S \to T} \circ \Phi_{T \to S}, \ I'_T = I_{T \to S} \circ \Phi_{S \to T} \quad (2)$$

Semantic distortion-consistent loss ($\mathcal{L}_{sem}$) enforces semantic consistency between the source image $I_S$ and its transformed pair $I_{S \to T}$ at the pixel-wise prediction level. The same constraint can be applied to between $I_T$ and $I_{T \to S}$. That is, an ideal deformation field generator $G$ should be able to generate $\Phi_{S \to T}$ and $\Phi_{T \to S}$, that reconstruct structural information from the pixel-wise prediction outputs of the distorted input images. With these semantic constraints, the relative distortion learning can be further improved by the segmentation model $M$; and also the segmentation adaptation can benefit from the improved quality of relative deformation fields. The loss function can be written as:

$$\mathcal{L}_{sem} = \|M(I_S) \circ \Phi_{S \to T} - M(I_{S \to T})\|_1 + \|M(I_T) \circ \Phi_{T \to S} - M(I_{T \to S})\|_1. \quad (3)$$

To minimize domain gaps in optical distortion between the transformed source images $I_{S \to T}$ and the target-domain images $I_T$, we introduce a distortion-aware discriminator $D_G$, which aims to discriminate distortion style between $I_T$ and $I_{S \to T}$. Similar to $G$, we again use the first-order image gradient, $\nabla I_T$ and $\nabla I_{S \to T}$, as input to $D_G$ along with the images. The loss function is defined as:

$$\mathcal{L}_{D_G} = \mathbb{E}_{I_S \sim \mathcal{S}, I_T \sim \mathcal{T}}[1 - D_G(I_S \circ \Phi_{S \to T}, \nabla(I_S \circ \Phi_{S \to T}))] + \mathbb{E}_{I_T \sim \mathcal{T}}[D_G(I_T, \nabla I_T)]. \quad (4)$$

We also calculate the adversarial loss using $D_G$:

$$\mathcal{L}_{adv\_G} = \mathbb{E}_{I_S \sim \mathcal{S}, I_T \sim \mathcal{T}}[D_G(I_S \circ \Phi_{S \to T}, \nabla(I_S \circ \Phi_{S \to T}))], \quad (5)$$

where the deformation generator $G$ tries to produce the relative deformation field to fool the discriminator $D_G$ in distinguishing distortion style.

Therefore, the total loss function for the relative distortion learning is defined as:

$$\mathcal{L}_{rdl} = \beta_1 \mathcal{L}_{recon} + \beta_2 \mathcal{L}_{sem} + \beta_3 \mathcal{L}_{adv\_G}, \quad (6)$$

where $\beta_1$, $\beta_2$, and $\beta_3$ are constants controlling the effect of corresponding losses.

## 3.2 Distortion-aware Adversarial Adaptation

Relative distortion learning aims to adapt optical distortion shifts at both pixel- and feature-level. Based on the adapted representations, we further apply adversarial segmentation adaptation. As our baseline adaptation methods, we take AdaptSeg [33] and AdvEnt [34], that have been extensively used in many prior work [26–28, 40] as for their based segmentation adaptation. Typically, the segmentation loss uses the cross-entropy for the transformed source image $I_{S \to T}$ and its corresponding label $Y_{S \to T}$:

$$\mathcal{L}_{seg} = -\sum_{h,w} \sum_{c \in C} Y_{S \to T}^{(h,w,c)} \log(M(I_{S \to T})^{(h,w,c)}), \quad (7)$$

where $Y_{S \to T}$ is obtained by transforming ground-truth annotations from the source domain by $Y_S \circ \Phi_{S \to T}$. Entropy minimization loss is further introduced by AdvEnt [34]:

$$\mathcal{L}_{ent} = \frac{-1}{\log(C)} \sum_{h,w} \sum_{c \in C} M(I_T)^{(h,w,c)} \log M(I_T)^{(h,w,c)}, \quad (8)$$

which tries to directly minimize pixel-wise entropies to enhance prediction certainty in the target domain. As a common practice, a domain discriminator $D_M$ is used to minimize the difference between the transformed source and target prediction probabilities:

$$\mathcal{L}_{D_M} = \mathbb{E}_{I_S \sim \mathcal{S}, I_T \sim \mathcal{T}}[1 - D_M(M(I_S \circ \Phi_{S \to T}))] + \mathbb{E}_{I_T \sim \mathcal{T}}[D_M(M(I_T))]. \quad (9)$$

The adversarial loss function for segmentation adaptation can be written as:

$$\mathcal{L}_{adv\_M} = \mathbb{E}_{I_T \sim \mathcal{T}}[D_M(1 - M(I_T))], \tag{10}$$

where the segmentation model $M$ is trained to fool the discriminator $D_M$.

Equipped with the relative distortion learning and the adversarial adaptation, we train the adaptive segmentation network with the following total loss:

$$\mathcal{L}_{all} = \mathcal{L}_{rdl} + \mathcal{L}_{adv\_M} + \mathcal{L}_{seg} + \gamma\mathcal{L}_{ent}, \tag{11}$$

where $\gamma$ is 0 for AdaptSeg [33] and 1 for AdvEnt [34].

## 4 Experimental Results

We present extensive experimental results to validate our distortion-aware domain adaptation (DaDA) framework for semantic segmentation in the presence of both visual and geometric domain shifts. For this, we formulate new domain adaptive segmentation benchmarks, *i.e.,* transferring from real-world or synthetic rectilinear images to real fisheye images.

### 4.1 Datasets

In the experiments, we first show evaluations of the model trained on real-world rectilinear images from the Cityscapes [8] and test the adapted model on real fisheye images from Woodscape [38] or our in-house fisheye driving dataset (FDD). Then, we introduce a more challenging adaptation task, where the model trained on synthetic dataset GTAV [29] is transferred to real fisheye images (Woodscape or FDD) without annotations. The Cityscapes dataset contains $2,975$ training images of high-quality driving scene with the resolution of $2048 \times 1024$. The GTAV dataset contains $24,966$ synthesized images with the resolution of $1914 \times 1052$. The Woodscape dataset consists of $8,234$ fisheye images with the resolution of $1280 \times 966$, where the images are captured by fisheye cameras ($190°$ F.O.V) looking at four different directions of the vehicle. We use front and rear camera scenes containing $4,023$ images in our experiments. The images are randomly split into a training set with $3,023$ images and a validation set with $1,000$ images. We report the results on 17 classes aligning mismatched classes between Cityscapes and GTAV; and Woodscape. Our in-house fisheye driving dataset (FDD) includes $3,897$ of fully annotated images with the resolution of $1920 \times 1080$ captured by fisheye cameras ($200°$ F.O.V) at front- and rear-side of the vehicle. We randomly pulled $974$ of validation images and remaining $2,923$ images are used for the training. For FDD, we use 12 classes, where incompatible classes in Cityscapes and GTAV are merged or excluded similar to Woodscape (see Appendix B.4 for detailed class information of Woodscape and FDD).

### 4.2 Experimental Details

**Diffeomorphic and Affine Transformation**

Fig.2 depicts an example of diffeomorphic and affine transformations applied to a source- and a target-domain image. First, both original target and source images are randomly cropped and resized to (a) and (b) followed by randomized horizontal flipping and photometric jittering. For the fine-grained diffeomorphic transformation, both target ($I_T$ in (a)) and source image ($I_S$ in (b)) are used to generate a transformed image $I_{S \to T}$ in (e) via the relative deformation field generator $G$. Additional global affine transformation is applied to $I_{S \to T}$ to generate an image in (f), which includes both fine-grained diffeomorphic and global affine deformations. For the affine transformation, we adopted RandAugment (RA) [9] excluding preceded photometric jittering. RA is one of the immediately probable

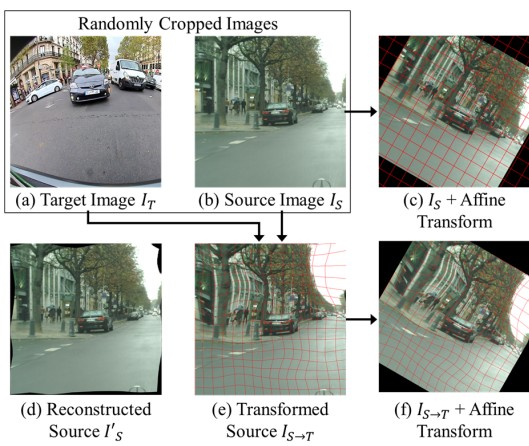

(a) Target Image $I_T$  (b) Source Image $I_S$  (c) $I_S$ + Affine Transform

(d) Reconstructed Source $I'_S$  (e) Transformed Source $I_{S \to T}$  (f) $I_{S \to T}$ + Affine Transform

Figure 2: **An example of diffeomorphic and affine transformations of training images.**

Table 1: **Comparisons with the baseline adaptation methods.** All the methods are based on DeepLab-V2 with ResNet-101 as the backbone for a fair comparison. Oracle performance (trained on target) for Woodscape is 67.02% and for FDD is 61.59%.

| Method | Cityscapes → Woodscape mIoU(%) | gain | GTAV → Woodscape mIoU(%) | gain | Cityscapes → FDD mIoU(%) | gain | GTAV → FDD mIoU(%) | gain |
|---|---|---|---|---|---|---|---|---|
| SourceOnly | 32.39 | | 29.32 | | 34.76 | | 32.13 | |
| AdaptSeg [33] | 46.33 | | 35.94 | | 39.07 | | 36.90 | |
| AdaptSeg+RA | 50.44 | +4.11 | 36.88 | +0.94 | 39.42 | +0.35 | 37.22 | +0.32 |
| AdaptSeg+RDL | 50.88 | +4.55 | 37.36 | +1.42 | **41.35** | **+2.28** | 39.29 | +2.39 |
| AdaptSeg+RA+RDL | **52.59** | **+6.26** | **37.73** | **+1.78** | 41.07 | +2.00 | **39.64** | **+2.74** |
| AdvEnt [34] | 45.26 | | 34.70 | | 38.87 | | 37.25 | |
| AdvEnt+RA | 50.60 | +5.34 | 36.64 | +1.94 | 41.58 | +2.71 | 38.75 | +1.50 |
| AdvEnt+RDL | 50.94 | +5.68 | 36.39 | +1.69 | **42.43** | **+3.56** | 39.93 | +2.68 |
| AdvEnt+RA+RDL | **52.64** | **+7.38** | **37.62** | **+2.92** | 42.32 | +3.45 | **40.87** | **+3.62** |

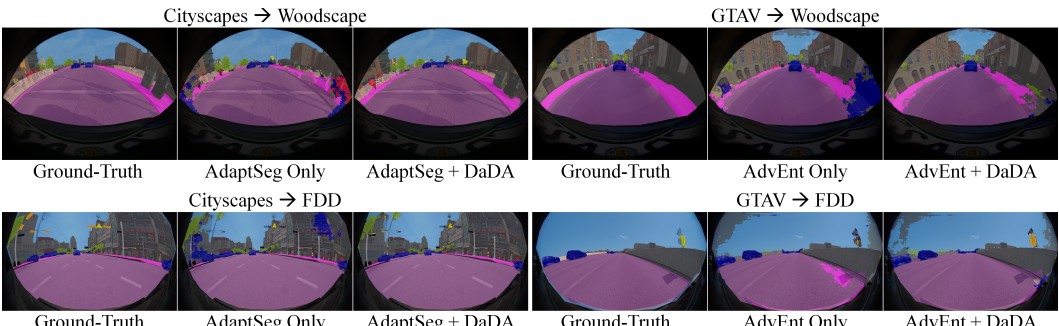

Figure 3: **Qualitative examples.** DaDA rectifies erroneous predictions, especially in the image periphery where severe optical distortions appear compared to the based adaptation methods.

and applicable augmentation methods including a series of affine transformations (*i.e.,* rotation, shear-x, shear-y, trans-x, and trans-y with 0.5 probability of application). For example, a rotation transformation is applied to the source image $I_S$ in (c). In our experiments, we tested transformed images generated by either one of affine-only (c), diffeomorphic-only (e), or both transformations (f).

**Implementation Details.** We trained all networks with the Adam [18] solver with a batch size of 4. The learning rate is $0.2 \times 10^{-5}$ for $M$ and $D_M$ and $0.1 \times 10^{-6}$ for $G$ and $D_G$. We set the weight factors of losses in Eq.(6) as: $\beta_1 = 100.0$, $\beta_2 = 10.0$, $\beta_3 = 10.0$ for Cityscapes → Woodscape (or FDD); and $\beta_1 = 100.0$, $\beta_2 = 1.0$, $\beta_3 = 100.0$ for GTAV → Woodscape (or FDD). Further details on the implementation as well as our hyperparameter selections are provided in Appendix A and B.

### 4.3 Comparisons with State-of-the-Art Methods

We first evaluate the effect of our distortion-aware adaptation framework when applied to the based adaptation methods: AdaptSeg [33] and AdvEnt [34]. We use DeepLab-V2 [6] with ResNet-101 backbone as the base semantic segmentation architecture $M$. We test the different spatial transformations (*+RDL*: applying a diffeomorphic transformation learned by RDL, *+RA*: applying an affine transformation via RandAugment [9], *+RA+RDL*: applying both diffeomorphic and affine transformations) on source images in the adaptation tasks.

**Woodscape as target domain (Tab.1).** Compared to the based adaptation methods, applying either one of fine-grained diffeomorphic (*+RDL*) or global affine transformations (*+RA*) achieves clear improvements on the all adaptation tasks. Remarkably, applying both fine-grained diffeomorphic and global affine transformations (*+RA+RDL*) achieves significant improvements of +7.38% and +2.92% on Cityscapes→Woodscape and GTAV→Woodscape tasks, respectively. Note that the performance improvement of the distortion-aware adaptation comes from object classes such as person (+20.07%), car (+14.49%), bus (+14.61%), and truck (+12.61%) as well as background such as sidewalk (+13.09%) (see class-wise iou(%) in Appendix B.4). Optical distortion gradually

Table 2: **Effect of DaDA on Self-Supervised Learning (SSL)**.

| SSL Method | +DaDA | Cityscapes → Woodscape | | GTAV → Woodscape | | Cityscapes → FDD | | GTAV → FDD | |
|---|---|---|---|---|---|---|---|---|---|
| | | mIoU(%) | gain | mIoU(%) | gain | mIoU(%) | gain | mIoU(%) | gain |
| IAST [26] | | 47.00 | | 38.83 | | 39.60 | | 37.47 | |
| | ✓ | 53.82 | +6.82 | **40.75** | **+1.92** | 44.46 | +4.86 | 40.06 | +2.59 |
| IntraDA [27] | | 48.92 | | 36.10 | | 40.36 | | 38.61 | |
| | ✓ | 53.24 | +4.32 | 39.85 | +3.75 | **45.28** | **+4.92** | **42.10** | **+3.49** |
| ProDA [40] | | 50.69 | | 34.44 | | 39.72 | | 35.97 | |
| | ✓ | **54.83** | **+4.14** | 35.75 | +1.31 | 42.14 | +2.42 | 37.09 | +1.12 |

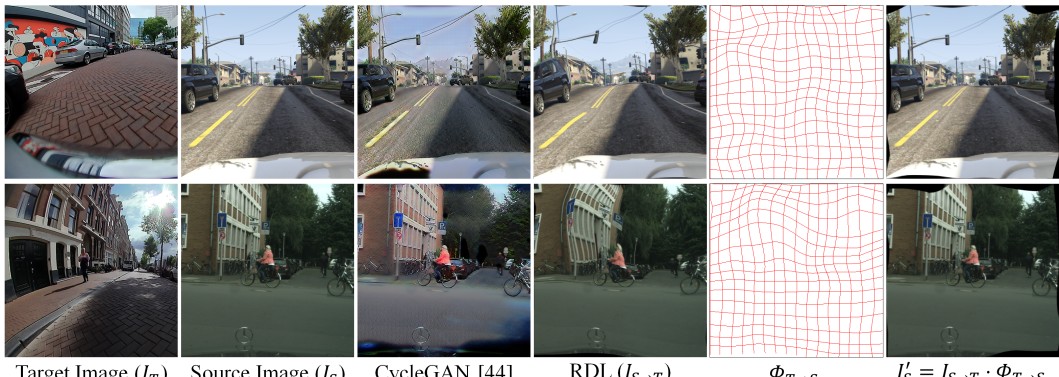

| Target Image ($I_T$) | Source Image ($I_S$) | CycleGAN [44] | RDL ($I_{S \to T}$) | $\Phi_{T \to S}$ | $I'_S = I_{S \to T} \cdot \Phi_{T \to S}$ |

Figure 4: **Comparisons of relative distortion learning (RDL) with an image-to-image translation method [42]** (Top row: GTAV → Woodscape, Bottom row: Cityscapes → Woodscape).

increases when objects and backgrounds appear closer to the image periphery. We hypothesize that our distortion adaptation method rectifies erroneous predictions in such distorted regions. These improvements are also confirmed in Fig.1-(b), Fig.3, and Fig.5 (See Section 4.4).

**FDD as target domain (Tab.1).** Here, the results are consistent with the previous adaptation tasks. Again, *+RA+RDL* clearly shows superior prediction performance to the based methods by up to +3.45% and +3.62% on Cityscapes→FDD and GTAV→FDD, respectively. Notably, *+RDL* contributes up to 3.56% (AdvEnt+RDL on Cityscapes → FDD) of improvements, which also consistently achieves higher performance gain than *+RA* (+2.71%). Such results are also echoed in the Cityscapes → CityscapesFishEye task in Appendix B.2. Note that *+RDL* always leads to improvements in segmentation adaptation, regardless of domain shift, throughout our experiments (*e.g.,* based methods vs. *+RDL*, and *+RA* vs. *+RA+RDL*) as presented in Tab.1 and Tab.7 in Appendix B.2. In contrast, the randomized affine augmentation (RA) leads to degraded segmentation adaptation results upon the geometric distributional shifts between source and target domains (*e.g.,* *+RDL* vs. *+RDL+RA* in Cityscapes → CityscapesFisheye and Cityscapes → FDD). Thus, we may conclude that our learnable diffeomorphic transformation (RDL) plays an important role in aligning the domain gap of geometric deformation.

**Relationship to SSL (Tab.2).** We evaluate the effect of our distortion-aware domain adaptation (DaDA), as a "*warm-up*" phase, for state-of-the-art adaptation methods using self-supervised learning: IAST [26], IntraDA [27], and ProDA [40]. In particular, we train the initial segmentation model with *+RA+RDL* distortion adaptation; and use AdaptSeg-based adaptation models for IAST [26] and ProDA [40]; and AdvEnt-based models for IntraDA [27]. Tab.2 shows the effectiveness of DaDA, where it further improves the self-supervised learning by providing higher-quality initial models and pseudo labels. DaDA attains up to +6.82% improvement against the baseline SSL methods. This implies that satisfactory distortion-aware adaptation cannot be achieved only by relying on SSL.

**Comparisons with Image-to-Image Translation.** We compare our relative distortion learning (RDL) with an existing image-to-image translation method CycleGAN [42]. Unpaired rectilinear source images (Cityscapes and GTAV) and fisheye target images (Woodscape) are given to both RDL and CycleGAN. The input images are randomly cropped and resized to $512 \times 512$. Fig.4 shows that CycleGAN fails to generate transformed images from source images mimicking the distortion style of

Table 3: **Ablation results on the distortion-aware losses.**

| Base Method | $+\mathcal{L}_{adv\_G}$ | $+\mathcal{L}_{sem}$ | $+\mathcal{L}_{recon}$ | Cityscapes $\rightarrow$ Woodscape | GTAV $\rightarrow$ Woodscape |
|---|---|---|---|---|---|
| | | | | 46.33 | 35.94 |
| | ✓ | | | 49.61 | 36.45 |
| AdaptSeg [33] | ✓ | ✓ | | 50.29 | 36.75 |
| | ✓ | | ✓ | 49.97 | 37.17 |
| | ✓ | ✓ | ✓ | **50.88** | **37.36** |
| | | | | 45.26 | 34.70 |
| | ✓ | | | 47.77 | 35.36 |
| AdvEnt [34] | ✓ | ✓ | | 49.22 | 35.77 |
| | ✓ | | ✓ | 50.32 | 36.11 |
| | ✓ | ✓ | ✓ | **50.94** | **36.39** |

target images. This is obvious to observe since CycleGAN-like existing translation approaches do not have devices (*e.g.,* diffeomorphic transformer) to geometrically transform source images. In contrast, RDL enables modeling relative distortion across domains and generates transformed source images alike the distortion style of target images. In Fig.4, buildings and vehicles are distorted replicating counterparts in target images via RDL, while CycleGAN focuses on translating texture and color of images. Fig.4 also demonstrates that RDL is able to reconstruct source images ($I_S$) from the distortion-translated images ($I_{S \rightarrow T}$) via an inverse relative deformation field $\Phi_{T \rightarrow S}$. Overall, these results imply that our distortion-aware losses ($\mathcal{L}_{recon}$, $\mathcal{L}_{sem}$) are effective in guiding the generator $G$ to produce convincing relative deformation fields.

## 4.4   Ablation Studies

To better understand the effect of our distortion-aware adaptation approach, we conduct ablation studies on the distortion-aware losses and the competence in predicting distorted image regions.

**Effect of Distortion-aware Losses (Tab.3).** We evaluate the effect of the distortion-aware losses on segmentation performance. Tab.3 depicts the improvement of prediction performance compared to the based methods, by adding individual or all of the proposed distortion-aware losses in Eq.(11). Only adding the adversarial loss ($+\mathcal{L}_{adv\_G}$) contributes to the adaptation performance up to $+3.28\%$ of improvements. Progressive introduction of the distortion-aware losses consistently improves prediction accuracy ($+\mathcal{L}_{sem}$, $+\mathcal{L}_{recon}$). Ultimately, utilizing all losses together with segmentation adaptation achieves $+5.68\%$ and $+1.69\%$ of improvements for Cityscapes $\rightarrow$ Woodscape and GTAV $\rightarrow$ Woodscape, respectively. Here we observe a relatively smaller improvement in GTAV $\rightarrow$ Woodscape task, which exhibits both severe visual and geometric distributional misalignment. We believe that introducing additional texture-aware translation methods along with our distortion adaptation approach might lead to further improvement in such a synthetic-to-real adaptation task. Overall, the relative distortion learning supervised by the distortion-aware losses effectively reduces domain shifts in optical distortion, and thus improves the prediction performance.

**Distortion-aware mIoU.** To quantitatively demonstrate the effectiveness of DaDA on predicting distorted image areas, we propose a distortion-aware mIoU(%) metric. The image coordinates are normalized to [-1.0,1.0] and we gradually mask label where $\sqrt{i^2 + j^2}$ of pixel at $(i, j)$ is smaller than a certain distance threshold ($dist$). In Fig.5 (bottom row), $dist = 0.0$ shows that the original label is used for mIoU(%) calculation, while $dist = 0.8$ indicates that large areas of undistorted regions are masked out in the label so that we evaluate the mod-

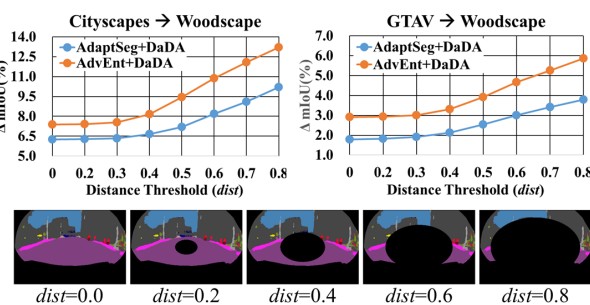

Figure 5: **Distortion-aware mIoU.**

els on distorted regions. Plots in Fig.5 (top row) show that the performance gain ($\Delta$mIoU) achieved by adding DaDA increases as $dist$ increases. This indicates that DaDA effectively addresses domain shifts in distortion and improves the prediction performance for the distorted image regions.

# 5 Discussion and Conclusion

In this paper, we proposed a novel distortion-aware domain adaptation (DaDA) framework that is capable of modeling domain shifts in geometric deformation based on a relative distortion learning (RDL) method. Besides, we demonstrated that our distortion adaptation approach further improves self-supervised learning by providing higher-quality initial models and pseudo labels. Extensive experimental results proved that our method minimizes domain shifts in optical distortion, and thus significantly improves the segmentation adaptation performance under unknown relative distortion across domains. In the future, we will investigate the interplay between the texture-oriented and the distortion-oriented domain shifts to further improve the unsupervised domain adaptation. While we first tackle adapting existing semantic segmentation models trained on rectilinear images to unlabeled fisheye images, various set-ups of domain adaptation tasks among distorted and rectilinear images can be further considered. For example, we may use distorted images as source and rectilinear images as a target, or both source and target domains include distorted images. Applying relative distortion learning to such extended adaptation tasks could be another interesting direction for future work. We hope our work provides a solid baseline and new perspectives on distortion-aware domain adaptation.

# 6 Disclosure of Funding

We received no third-party funding for this work.

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
