# [Supplementary Material]
# DaDA: Distortion-aware Domain Adaptation for Unsupervised Semantic Segmentation

**Sujin Jang**
Samsung Advanced Institute of Technology
`s.steve.jang@samsung.com`

**Joohan Na**
Samsung Advanced Institute of Technology
`joohan.na@samsung.com`

**Dokwan Oh**
Samsung Advanced Institute of Technology
`dokwan.oh@samsung.com`

In this supplementary material, we provide further details of training and implementation of our method (Appendix A). Then, we present additional experimental results including the sensitivity of our distortion-aware losses, effect of disentangled geometric distortion on the adaptation task, distortion-aware mIoU(%), and class-wise segmentation performance (Appendix B). Finally, we provide further qualitative results where DaDA clearly shows improved prediction performance in the presence of unknown radial distortion across domains (Appendix C).

## A    Further Implementation Details

### A.1    Network Architectures

We adopted DeepLab-V2 [1] with ResNet-101 backbone as the base semantic segmentation architecture $M$. We used the ImageNet [2] pre-trained weights to initialize the backbone. For the discriminator $D_M$, we followed the architectures of AdaptSeg [10] and AdvEnt [11]. For our relative distortion learning (RDL), we designed the deformation field generator $G$ with 5 fully-convolutional layers with kernel 4×4 and stride of 2, where each layer has 32, 64, 128, 256, and 512 channels, respectively. Each convolution layer is followed by an instance norm and a leaky ReLU with 0.2 of the negative slope parameter. Subsequently, two additional convolutional layers with kernel 3×3 are added to extract the flow field $u$, where each convolutional layer has 256 and 2 channels; and is followed by a leaky ReLU and a hyperbolic tangent activation, respectively. For the distortion-aware discriminator $D_G$, we used the same architecture as the one used in PatchGAN [4] to classify distortion style. In training $D_G$, we applied the one-sided label smoothing [8] where the positive labels are smoothed to 0.9 instead of 1.0. Such a smoothing technique forbids the discriminator from overwhelming the generator $G$ by penalizing the overconfidence of the discriminator [3].

### A.2    Training Details

The training procedure of our distortion-aware domain adaptation is summarized in Algorithm 1. To implement our relative distortion learning jointly with the segmentation adaptation, we added the proposed distortion-aware discriminator and deformation field generator $G$ to the based adaptation implementations. Our baseline adaptation methods, AdaptSeg [10] and AdvEnt [11] were implemented using the authors' official releases. We also directly used the official release of IntraDA [7], IAST [6], and ProDA [12] following the training procedure of self-supervised learning phases. Note that only ProDA [12] requires modified ASPP (astrous spatial pyramid pooling) layer of DeepLab-V2, and we followed its official implementation.

36th Conference on Neural Information Processing Systems (NeurIPS 2022).

**Algorithm 1: Distortion-aware Domain Adaptation (DaDA)**

**Input:** training dataset: $(I_S, Y_S, I_T)$; pre-trained segmentation model with source-only dataset: $M_0$
**Output:** adapted segmentation model: $M^*$

1   Initialize $M$ with $M_0$;
2   **for** $m \leftarrow 0$ **to** epochs **do**
3      **for** $i \leftarrow 0$ **to** $\text{len}(I_T)$ **do**
4         Get source images $I_S^{(i)}$ and target images $I_T^{(i)}$;
5         Generate forward and inverse deformation fields: $(\Phi_{S\rightarrow T}^{(i)}, \Phi_{T\rightarrow S}^{(i)}) \leftarrow \text{G}(I_S^{(i)}, I_T^{(i)}, \nabla I_S^{(i)}, \nabla I_T^{(i)})$;
6         Generate transformed images and labels: $I_{S\rightarrow T}^{(i)} \leftarrow I_S^{(i)} \circ \Phi_{S\rightarrow T}^{(i)}$, $I_{T\rightarrow S}^{(i)} \leftarrow I_T^{(i)} \circ \Phi_{T\rightarrow S}^{(i)}$,
         $Y_{S\rightarrow T}^{(i)} \leftarrow Y_S^{(i)} \circ \Phi_{S\rightarrow T}^{(i)}$;
7         Update the model $M$ with $(I_{S\rightarrow T}^{(i)}, Y_{S\rightarrow T}^{(i)}, I_T^{(i)})$ using losses $\mathcal{L}_{seg}$, $\mathcal{L}_{ent}$ and $\mathcal{L}_{adv\_M}$;
8         Generate reconstructed source and target images: $I_S'^{(i)} \leftarrow I_{S\rightarrow T}^{(i)} \circ \Phi_{T\rightarrow S}^{(i)}$,
         $I_T'^{(i)} \leftarrow I_{T\rightarrow S}^{(i)} \circ \Phi_{S\rightarrow T}^{(i)}$;
9         Update the model $G$ with $(I_S^{(i)}, I_{S\rightarrow T}^{(i)}, I_S'^{(i)}, I_T'^{(i)}, \Phi_{S\rightarrow T}^{(i)}, \Phi_{T\rightarrow S}^{(i)}, M)$ using losses $\mathcal{L}_{recon}$, $\mathcal{L}_{sem}$,
         and $\mathcal{L}_{adv\_G}$
10     $M^* \leftarrow M$

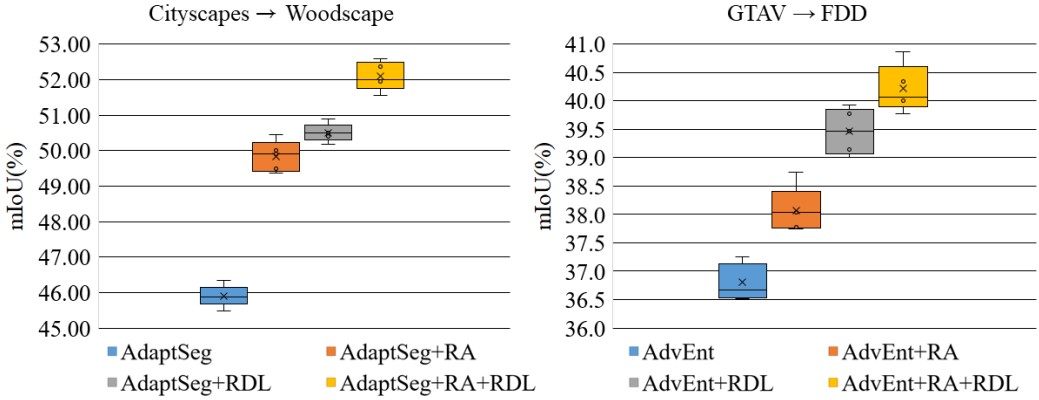

Figure 6: **A summary of performance statistics of adaptation models under different random seeds.**

Our DaDA framework typically needs 15~17K iterations with a batch size of 4 until convergence (*i.e.,* excluding source-only and self-supervised learning). This is about 20 epochs of the target dataset Woodscape and FDD. Both source and target images are randomly cropped and resized along with randomized horizontal flipping and photometric jittering. Specifically, the crop height is randomly selected from [341,950] for GTAV, [341,1000] for Cityscapes, [386,942] for Woodscape, and [430,1503] for FDD; and the images are resized to $768 \times 768$. At the test stage, we used the validation set of Woodscape and FDD with their original size $1280 \times 966$ and $1920 \times 1080$, respectively. We also did not use multi-scale inputs for evaluation. We conducted our experiments using PyTorch v1.8.0, CUDA v11.1, CuDNN v8.0.5; and all experiments were done on a single NVIDIA A100 GPU.

To test the stability and consistency of the adaptation methods over random seeds, we ran experiments with exactly the same configurations and hyperparameters, but with five different random seeds. Tab.4 and Fig.6 summarize the prediction performance of the model with different random seeds. Here we note clear improvements by our proposed method in all adaptation tasks in Fig.6.

Table 4: **Performance statistics obtained from models trained with five different random seeds.**

| Method | Task | avg. | std. |
|---|---|---|---|
| AdaptSeg [10] | Cityscapes→Woodscape | 45.90 | 0.301 |
| AdaptSeg+RA | Cityscapes→Woodscape | 49.84 | 0.431 |
| AdaptSeg+RDL | Cityscapes→Woodscape | 50.50 | 0.253 |
| AdaptSeg+RA+RDL | Cityscapes→Woodscape | 52.09 | 0.403 |
| AdvEnt [11] | GTAV→FDD | 36.80 | 0.318 |
| AdvEnt+RA | GTAV→FDD | 38.07 | 0.405 |
| AdvEnt+RDL | GTAV→FDD | 39.46 | 0.394 |
| AdvEnt+RA+RDL | GTAV→FDD | 40.22 | 0.418 |

Table 5: **The influence of $\beta_1$ ($\mathcal{L}_{recon}$) and $\beta_2$ ($\mathcal{L}_{sem}$) in relative distortion learning.**

**Cityscapes → Woodscape**

| $\beta_2 \downarrow$ $\quad$ $\beta_1 \rightarrow$ | 1.0 | 10.0 | 100.0 | 150.0 |
|---|---|---|---|---|
| 1.0 | 49.69 | 50.71 | 50.07 | 49.84 |
| 10.0 | 49.68 | 50.74 | **50.88** | 49.68 |
| 100.0 | 50.70 | 50.64 | 50.50 | 50.43 |
| 150.0 | 50.68 | 50.54 | 50.10 | 50.61 |

**GTAV → Woodscape**

| $\beta_2 \downarrow$ $\quad$ $\beta_1 \rightarrow$ | 1.0 | 10.0 | 100.0 | 150.0 |
|---|---|---|---|---|
| 0.1 | 36.11 | 36.35 | 36.87 | 36.32 |
| 1.0 | 36.47 | 36.58 | **37.36** | 36.81 |
| 10.0 | 36.21 | 36.50 | 37.21 | 36.98 |
| 100.0 | 36.36 | 36.96 | 36.82 | 36.16 |

Table 6: **The influence of $\beta_3$ for the adversarial loss ($\mathcal{L}_{adv\_G}$) in relative distortion learning.**

**Cityscapes → Woodscape**

| $\beta_3$ | 1.0 | 10.0 | 100.0 | 150.0 |
|---|---|---|---|---|
| mIoU(%) | 50.04 | **50.88** | 50.75 | 50.02 |

**GTAV → Woodscape**

| $\beta_3$ | 1.0 | 10.0 | 100.0 | 150.0 |
|---|---|---|---|---|
| mIoU(%) | 36.81 | 36.85 | **37.36** | 36.93 |

# B  Additional Experimental Results

## B.1  Hyperparameter Search and Sensitivity

To select hyperparameters for our distortion-aware loss functions, we first performed experiments with a few reasonable choices of the weight values $\beta_1(\mathcal{L}_{recon})$ and $\beta_2(\mathcal{L}_{sem})$. Here, we report the results in mIoU(%) from AdaptSeg+RDL on both Cityscapes→Woodscape and GTAV→Woodscape tasks. Tab.5 shows that our add-on adaptation method (+RDL) maintains strong accuracy under different settings of $\beta_1$ and $\beta_2$. We also investigated the sensitivity of the weight value $\beta_3$ of the adversarial loss $\mathcal{L}_{adv\_G}$. Tab.6 shows that RDL is not sensitive to $\beta_3$ in a vast range.

## B.2  Effect of Disentangled Geometric Distortion

To clarify the effect of geometric distortion on the adaptation tasks, we performed an additional experiment where the geometric distortion is isolated from other factors in distributional shifts (*e.g.,* visual domain gaps). To be more specific, we took the Cityscapes dataset as source and its distorted counterpart as target (CityscapesFishEye) including fisheye-like images similar to $I_T$ in Fig.1 of the manuscript. The distorted images are synthetically generated based on the equidistance fisheye

Table 7: **Comparisons with the baseline adaptation methods on Cityscapes → CityscapesFish-Eye adaptation task.** All the methods are based on DeepLab-V2 with ResNet-101 as the backbone for a fair comparison.

| Method | mIoU(%) | gain |
|---|---|---|
| Oracle (trained on target) | 69.82 | |
| Source Only (trained on source) | 35.69 | |
| AdatSeg [10] | 47.16 | |
| AdaptSeg+**RDL** | 57.89 | +10.73 |
| Adaptseg+RA | 54.02 | |
| AdaptSeg+RA+**RDL** | 55.58 | +1.56 |
| AdvEnt [7] | 46.67 | |
| AdvEnt+**RDL** | 57.04 | +10.37 |
| AdvEnt+RA | 54.55 | |
| AdvEnt+RA+**RDL** | 55.82 | +1.27 |

Table 8: **Performance gain achieved by adding DaDA increases as $dist$ increases.**

| Cityscapes → Woodscape | | | | | | | | | |
|---|---|---|---|---|---|---|---|---|---|
| Method | dist=0.0 | dist=0.2 | dist=0.4 | dist=0.6 | dist=0.8 | gain@0.0 | gain@0.2 | gain@0.4 | gain@0.6 | gain@0.8 |
| AdaptSeg [10] | 46.33 | 46.27 | 46.11 | 43.89 | 38.22 | | | | | |
| AdaptSeg+DaDA | 52.59 | 52.55 | 52.78 | 52.08 | 48.42 | +6.26 | +6.28 | +6.67 | +8.19 | +10.20 |
| AdvEnt [7] | 45.26 | 45.19 | 44.97 | 42.54 | 37.44 | | | | | |
| AdvEnt+DaDA | 52.64 | 52.60 | 53.14 | 53.41 | 50.65 | +7.38 | +7.41 | +8.17 | +10.87 | +13.21 |
| GTAV → Woodscape | | | | | | | | | |
| Method | dist=0.0 | dist=0.2 | dist=0.4 | dist=0.6 | dist=0.8 | gain@0.0 | gain@0.2 | gain@0.4 | gain@0.6 | gain@0.8 |
| AdaptSeg [10] | 35.94 | 35.92 | 35.68 | 33.95 | 30.46 | | | | | |
| AdaptSeg+DaDA | 37.73 | 37.74 | 37.80 | 36.95 | 34.25 | +1.78 | +1.82 | +2.12 | +3.00 | +3.79 |
| AdvEnt [7] | 34.70 | 34.67 | 34.54 | 32.94 | 28.94 | | | | | |
| AdvEnt+DaDA | 37.62 | 37.61 | 37.85 | 37.60 | 34.81 | +2.92 | +2.94 | +3.31 | +4.66 | +5.87 |

camera projection model [5]. Results from Tab.7 clearly show that our relative distortion learning (RDL) contributes to significant improvements in the adaptation performance up to +10.73% when only geometric distortion is presented in the distributional shifts. This is obvious to observe since the baseline methods (*i.e.,* AdaptSeg [10], AdvEnt [11]) do not consider the geometric distortion in domain shifts while our approach features distortion-aware adaptation based on relative distortion learning (RDL). Remarkably, +RDL achieves the largest gain over the based method and such results are echoed in the Cityscapes → FDD task in Tab.1 of the manuscript.

## B.3 Distortion-aware mIoU(%)

To quantitatively demonstrate the effectiveness of DaDA on the performance of predicting distorted image areas, we proposed a distortion-aware mIoU(%) metric in the main manuscript (see Section 4.4 and Fig.5.). Here we provide further details of the performance gain achieved by adding DaDA in Tab.8. We observed that performance gain achieved by adding DaDA increases as $dist$ increases. This indicates that DaDA effectively addresses domain shifts in distortion and improves the prediction performance for the distorted image regions. Note that these results are echoed in Fig.7 and Fig.8 as well as the qualitative results in the main manuscript.

## B.4 Class-wise Semantic Segmentation Performance

In Tab.9 and Tab.10, we present class-wise performance of the segmentation adaptation methods. For Wooesapce, unlike common 19-class definition in semantic segmentation, we report the results on 17 classes since there are mismatches between Cityscapes and GTAV; and Woodscape (*i.e.,* merged "building" and "wall" classes to "construction" class; and "vegetation" and "terrain" classes to "nature"

Table 9: **Comparisons with the baseline adaptation methods on Cityscapes → Woodscape and GTAV → Woodscape tasks.** We report mIoUs(%) with respect to 17 classes. All the methods are based on DeepLab-V2 with ResNet-101 as the backbone for a fair comparison.

| Method | road | side. | const. | fence | pole | light | sign | nature | sky | person | rider | car | truck | bus | train | motor | bike | mIoU | gain |
|---|---|---|---|---|---|---|---|---|---|---|---|---|---|---|---|---|---|---|---|
| Oracle | 96.62 | 72.91 | 88.69 | 53.19 | 28.67 | 34.70 | 39.22 | 86.48 | 95.99 | 62.11 | 55.88 | 89.99 | 64.86 | 59.94 | 86.68 | 60.92 | 62.50 | 67.02 | |
| **Cityscapes → Woodscape** | | | | | | | | | | | | | | | | | | | |
| Source Only | 82.21 | 11.02 | 70.19 | 4.64 | 8.90 | 15.48 | 11.12 | 62.96 | 82.82 | 28.24 | 20.99 | 32.76 | 18.53 | 18.58 | 15.19 | 31.72 | 35.29 | 32.39 | |
| AdaptSeg[10] | 89.70 | 39.58 | 79.14 | 17.13 | 17.67 | 19.61 | 21.74 | 79.72 | 91.57 | 38.15 | 41.31 | 53.54 | 36.44 | 32.90 | 43.98 | 38.71 | 46.68 | 46.33 | |
| AdaptSeg+RA | 90.56 | 45.45 | 79.76 | 24.33 | 22.42 | 23.32 | 28.35 | 79.34 | 88.01 | 48.01 | 41.76 | 59.67 | 48.45 | 37.05 | 49.14 | 43.39 | 48.46 | 50.44 | +4.11 |
| AdaptSeg+RDL | 90.77 | 35.28 | 80.47 | 21.44 | 21.87 | 23.93 | 25.31 | 80.71 | 90.77 | 45.19 | 40.39 | 70.15 | 49.26 | 40.17 | 55.47 | 43.75 | 50.05 | 50.88 | +4.55 |
| AdaptSeg+RA+RDL | 92.11 | 44.79 | 80.91 | 19.45 | 21.69 | 22.88 | 26.90 | 80.23 | 90.72 | 44.30 | 40.53 | 76.73 | 54.68 | 47.42 | 55.83 | 44.51 | 50.41 | **52.59** | **+6.26** |
| AdvEnt[11] | 89.16 | 31.96 | 77.43 | 24.02 | 18.18 | 19.87 | 16.15 | 77.03 | 90.56 | 28.44 | 33.28 | 58.70 | 43.41 | 39.22 | 49.93 | 33.08 | 39.03 | 45.26 | |
| AdvEnt+RA | 91.18 | 49.67 | 80.53 | 26.41 | 22.49 | 23.12 | 20.18 | 78.73 | 90.58 | 43.34 | 38.81 | 67.07 | 48.65 | 44.61 | 48.09 | 41.09 | 45.69 | 50.60 | +5.34 |
| AdvEnt+RDL | 91.32 | 38.48 | 81.23 | 21.86 | 21.55 | 22.94 | 16.23 | 78.40 | 90.79 | 47.26 | 40.28 | 74.36 | 51.33 | 52.23 | 55.49 | 38.18 | 44.02 | 50.94 | +5.68 |
| AdvEnt+RA+RDL | 91.91 | 45.05 | 81.37 | 22.68 | 21.81 | 22.71 | 22.10 | 78.61 | 89.88 | 48.51 | 40.13 | 73.19 | 56.02 | 53.83 | 64.21 | 37.89 | 44.90 | **52.64** | **+7.38** |
| **GTAV → Woodscape** | | | | | | | | | | | | | | | | | | | |
| Source Only | 68.89 | 11.30 | 66.24 | 16.18 | 10.34 | 17.03 | 12.19 | 58.34 | 67.87 | 38.02 | 15.78 | 36.70 | 16.74 | 6.30 | 10.76 | 25.65 | 20.07 | 29.32 | |
| AdaptSeg[10] | 83.32 | 18.90 | 77.44 | 17.37 | 15.80 | 20.68 | 12.42 | 68.46 | 84.88 | 41.63 | 19.77 | 65.66 | 29.63 | 15.82 | 7.73 | 21.10 | 10.31 | 35.94 | |
| AdaptSeg+RA | 83.96 | 26.88 | 77.90 | 19.58 | 18.19 | 21.72 | 14.70 | 71.31 | 85.75 | 41.62 | 16.81 | 64.31 | 26.20 | 16.19 | 10.65 | 30.40 | 7.04 | 36.88 | +0.94 |
| AdaptSeg+RDL | 83.87 | 20.00 | 76.90 | 17.29 | 14.53 | 21.01 | 13.61 | 71.35 | 84.06 | 42.68 | 15.73 | 68.30 | 27.35 | 14.72 | 7.10 | 31.40 | 25.17 | 37.36 | +1.42 |
| AdaptSeg+RA+RDL | 84.31 | 29.42 | 77.52 | 25.88 | 16.49 | 22.36 | 14.86 | 68.73 | 83.46 | 40.81 | 18.03 | 67.90 | 29.75 | 19.24 | 9.79 | 31.33 | 13.50 | **37.72** | **+1.78** |
| AdvEnt[11] | 77.41 | 18.85 | 73.56 | 18.51 | 14.42 | 19.96 | 11.06 | 65.39 | 80.10 | 39.97 | 12.64 | 55.28 | 31.86 | 22.01 | 10.45 | 26.83 | 11.63 | 34.70 | |
| AdvEnt+RA | 80.68 | 23.54 | 76.48 | 21.25 | 17.68 | 19.24 | 12.41 | 66.51 | 82.01 | 40.50 | 10.24 | 66.82 | 35.69 | 25.82 | 16.53 | 30.42 | 17.11 | 36.64 | +1.94 |
| AdvEnt+RDL | 83.97 | 21.51 | 77.28 | 20.09 | 15.22 | 18.35 | 10.21 | 65.75 | 83.56 | 40.59 | 11.48 | 67.69 | 38.65 | 25.48 | 19.28 | 27.49 | 17.03 | 36.39 | +1.69 |
| AdvEnt+RA+RDL | 82.53 | 18.70 | 72.55 | 15.82 | 18.13 | 20.56 | 14.34 | 65.29 | 80.07 | 41.72 | 21.17 | 70.24 | 38.55 | 28.19 | 12.50 | 29.43 | 19.80 | **37.62** | **+2.92** |

Table 10: **Comparisons with the baseline adaptation methods on Cityscapes → FDD and GTAV → FDD tasks.** We report mIoUs(%) with respect to 12 classes. All the methods are based on DeepLab-V2 with ResNet-101 as the backbone for a fair comparison.

| Method | road | side. | const. | fence | pole | light | sign | nature | sky | person | rider | car | mIoU | gain |
|---|---|---|---|---|---|---|---|---|---|---|---|---|---|---|
| Oracle | 99.02 | 59.44 | 86.53 | 87.91 | 24.99 | 39.48 | 29.05 | 85.70 | 96.68 | 40.09 | 61.32 | 88.91 | 61.59 | |
| **Cityscapes → FDD** | | | | | | | | | | | | | | |
| Source Only | 95.16 | 10.51 | 53.46 | 1.49 | 11.84 | 3.28 | 3.47 | 67.04 | 93.87 | 3.33 | 15.21 | 58.44 | 34.76 | |
| AdaptSeg[10] | 94.23 | 6.09 | 57.41 | 3.02 | 12.71 | 11.49 | 7.63 | 73.48 | 94.41 | 14.84 | 26.88 | 66.64 | 39.07 | |
| AdaptSeg+RA | 94.60 | 8.41 | 54.79 | 5.14 | 11.00 | 13.27 | 9.00 | 73.71 | 92.01 | 21.06 | 21.81 | 68.23 | 39.42 | +0.35 |
| AdaptSeg+RDL | 94.86 | 9.77 | 55.84 | 4.39 | 11.09 | 14.73 | 6.71 | 75.07 | 92.94 | 21.48 | 40.28 | 69.04 | **41.35** | **+2.28** |
| AdaptSeg+RA+RDL | 95.58 | 17.33 | 56.87 | 4.28 | 12.45 | 10.68 | 6.40 | 74.18 | 93.16 | 19.57 | 34.32 | 68.06 | 41.07 | +2.00 |
| AdvEnt[11] | 95.06 | 7.92 | 53.99 | 2.47 | 11.80 | 11.75 | 4.33 | 73.73 | 93.03 | 6.86 | 33.53 | 71.96 | 38.87 | |
| AdvEnt+RA | 96.84 | 16.57 | 56.21 | 2.41 | 11.44 | 16.10 | 4.56 | 73.65 | 93.79 | 13.70 | 43.60 | 70.11 | 41.58 | +2.71 |
| AdvEnt+RDL | 96.87 | 15.94 | 58.94 | 1.15 | 9.48 | 14.22 | 5.05 | 74.31 | 92.91 | 16.88 | 49.76 | 73.64 | **42.43** | **+3.56** |
| AdvEnt+RA+RDL | 96.27 | 20.37 | 58.81 | 1.41 | 10.72 | 14.44 | 5.53 | 71.3 | 93.86 | 24.68 | 39.80 | 70.61 | 42.32 | +3.45 |
| **GTAV → FDD** | | | | | | | | | | | | | | |
| Source Only | 62.82 | 1.27 | 37.99 | 0.78 | 12.27 | 4.90 | 1.97 | 66.79 | 85.14 | 21.18 | 21.45 | 61.99 | 32.13 | |
| AdaptSeg[10] | 89.35 | 3.30 | 45.34 | 0.54 | 13.36 | 10.06 | 5.26 | 69.06 | 85.02 | 26.63 | 32.34 | 62.53 | 36.90 | |
| AdaptSeg+RA | 89.34 | 5.65 | 42.92 | 1.54 | 11.97 | 12.26 | 5.61 | 68.34 | 84.21 | 25.08 | 36.23 | 63.52 | 37.22 | +0.32 |
| AdaptSeg+RDL | 90.84 | 6.87 | 50.12 | 1.20 | 13.01 | 12.71 | 6.65 | 67.72 | 89.48 | 27.52 | 44.05 | 61.33 | 39.29 | +2.39 |
| AdaptSeg+RA+RDL | 95.33 | 14.20 | 56.05 | 3.50 | 12.89 | 11.94 | 6.42 | 73.67 | 93.57 | 20.26 | 30.36 | 67.72 | **39.64** | **+2.74** |
| AdvEnt[11] | 89.16 | 3.98 | 45.92 | 0.97 | 12.26 | 11.35 | 2.87 | 70.71 | 86.64 | 25.84 | 31.00 | 66.26 | 37.25 | |
| AdvEnt+RA | 89.87 | 4.77 | 42.27 | 1.61 | 12.86 | 11.95 | 5.37 | 71.05 | 85.80 | 28.20 | 45.72 | 65.50 | 38.75 | +1.50 |
| AdvEnt+RDL | 92.14 | 4.85 | 54.32 | 0.94 | 13.61 | 12.67 | 4.53 | 69.88 | 91.33 | 30.60 | 37.05 | 67.28 | 39.93 | +2.68 |
| AdvEnt+RA+RDL | 93.91 | 7.42 | 44.17 | 1.44 | 13.38 | 14.87 | 4.02 | 72.02 | 87.05 | 30.70 | 54.17 | 67.24 | **40.87** | **+3.62** |

class). For FDD, we use 12 classes, where incompatible classes are merged or excluded similar to Woodscape (*i.e.,* merged "construction" and "nature"; "truck" and "bus" into "car").

## C Additional Qualitative Results

Here we provide further qualitative comparisons of our distortion-aware domain adaptation (DaDA) with the based adaptation methods. In Fig.7 and 8, we observed clear qualitative improvements over the based adaptation methods. Optical distortion gradually increases towards the image periphery, and such distortion makes the based adaptation methods fail in large areas of objects and background far from the image center. Fig.7 and 8 show that DaDA fixes such false predictions of large areas of background (*e.g., road* and *sky*) close to the image periphery. DaDA also improves pixel-wise prediction of objects under severe distortion (*e.g., car*, *person*, and *bus*) by diminishing domain gaps at the input- and the output feature-level. Fig.9 also demonstrates that DaDA shows stronger and finer boundary of class-wise activation visualizations [9] under severe radial distortion across domains. In addition, we observe that DaDA rectifies large areas of false class-wise activation (*e.g.,* false activation of *person* on the ground, around target classes, and nowhere).

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

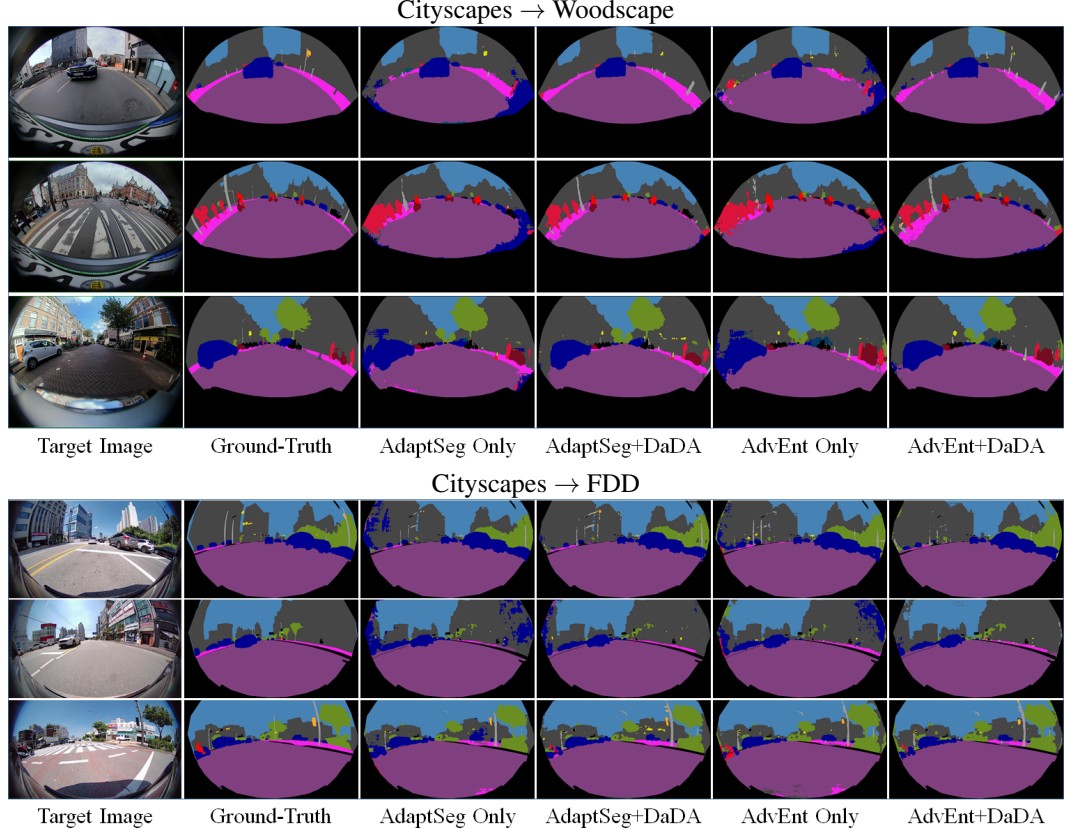

Figure 7: **Qualitative Results.** Each row tests Woodscape or FDD images along with corresponding Ground-Truth and presents prediction results from the baseline adaptation and our distortion-aware adaptation approach (DaDA).

GTAV → Woodscape

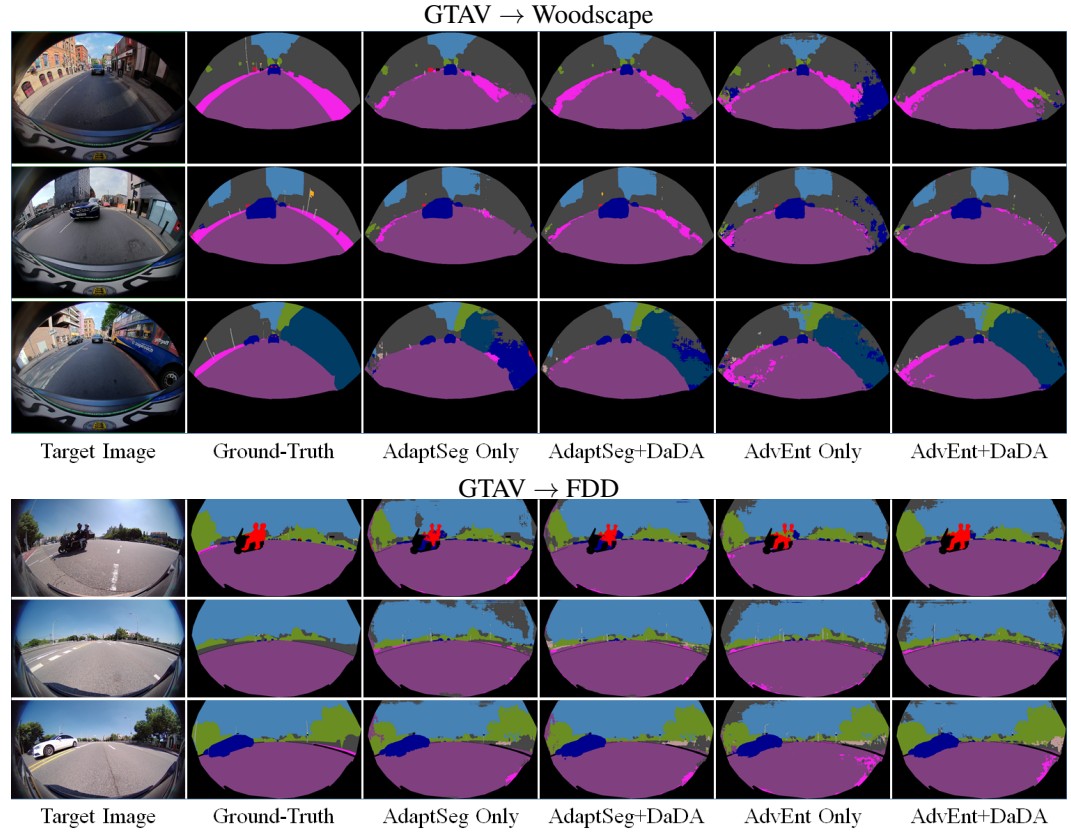

Figure 8: **Qualitative Results.** Each row tests Woodscape or FDD images along with corresponding Ground-Truth and presents prediction results from the baseline adaptation and our distortion-aware adaptation approach (DaDA).

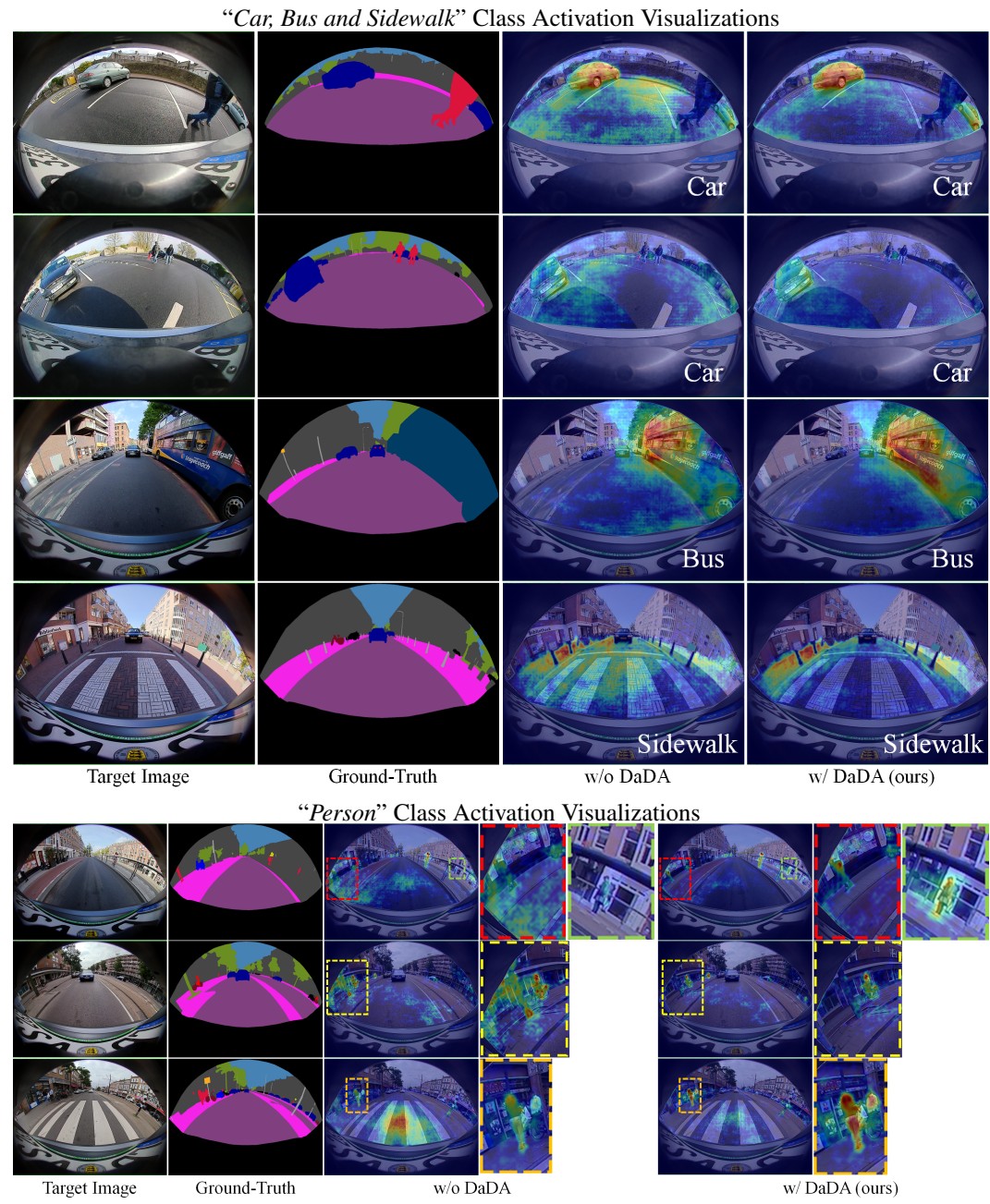

Figure 9: **Class-wise activation visualization using Grad-Cam [9].**