# OpenReview forum: "DaDA: Distortion-aware Domain Adaptation for Unsupervised Semantic Segmentation"
_NeurIPS.cc/2022/Conference — NeurIPS 2022 Accept_

### Official Review · Reviewer_ihqK · 2022-07-03

**Rating:** 6
**Confidence:** 4
**Soundness:** 2 fair
**Presentation:** 3 good
**Contribution:** 2 fair

**Summary:**

The author proposed a distortion-aware domain adaptation (DaDA) framework that is capable of modeling domain shifts in geometric deformation based on a relative distortion learning (RDL) method.

The proposed method tackles the task of unsupervised domain adaptation for semantic image segmentation where unknown optical distortion exists between the source and target images.

Adequate experimental results also prove the validity of the proposed method。

**Questions:**

1.The figures in the text are far apart from where the figures are mentioned in the body text. It is suggested that the author shorten the corresponding distance to facilitate the reader’s comparative reading.

2.Why are the edges of the reconstructed source domain image not neat? Such as the “d” part in figure2.

3.In the "Experimental Details" section, the performance of both the “AdaptSeg+RA+RDL” and “AdvEnt+RA+RDL” methods decreases compared to the RA removal method when cityscapes is the source domain dataset and FDD is the target domain dataset. The authors need an explanation for this experimental result.

4.Are the references in [17] and [18] duplicated?

5.The format of references is not uniform. For example, the format of the reference in [28] differs from the rest of the literature.

6.Most of the references are earlier than 2020. It is recommended to cite more papers in related fields after 2021.

**Ethics Review Area:**

["I don’t know"]

**Limitations:**

The authors adequately address the limitations of their work and the potential negative social impact.

**Strengths And Weaknesses:**

1.Strengths
This paper proposes a novel domain adaptation method for unsupervised semantic segmentation. And it combines geometric and optical distortion in domain shift. And the extensive experimental results highlight the effectiveness of our approach over state-of-the-art methods under unknown relative distortion across domains.
2.Weaknesses
See the “Questions” part.

---

> ### Author Response · Authors · 2022-08-02
> **Answers to `ihqk'**
>
> We thank the reviewer ‘ihqk’ for the informative and carefully written comments, and hope you find our answers helpful to resolve your questions.
>
> * **A1.** Thanks for your suggestion and we will try to shorten the distance between figures and corresponding paragraphs in the final manuscript.
>
> * **A2.** Such irregular edges of the reconstructed images (Fig.2-(d) in the manuscript) can be seen as the artifacts of the diffeomorphic transformation.
> We may consider adding a constraint to make the boundary of the reconstructed image to be aligned with the edge.
> However, it is unknown whether such constraint affects the performance of the segmentation adaptation and we would like to investigate this in future work.
>
> * **A3.**
> As 'ihqk' pointed out, we found that the randomized affine augmentation via RA is not always beneficial to the segmentation adaptation when it is combined with our relative distortion learning (RDL).
> As we discussed in Line 282-286 of the manuscript, the effectiveness of RA may depend on the geometric distributional shifts between source and target domains.
> For example, +RDL+RA shows degraded segmentation adaptation performance than +RDL in   *Cityscapes $\rightarrow$ FDD* in Tab.1 of the manuscript, and *Cityscapes $\rightarrow$ CityscapesFisheye* in above comment for the reviewer 'QLQF' ([link]( https://openreview.net/forum?id=6RoAxmwj0L2&noteId=ew14s2TDnI "Title")). \
> In contrast, RDL always leads to improvements in segmentation adaptation, regardless of pairs of datasets, throughout our experiments (e.g., based methods vs. +RDL, and +RA vs. +RA+RDL) as presented in Tab.1 of the manuscript and above result table ([link]( https://openreview.net/forum?id=6RoAxmwj0L2&noteId=ew14s2TDnI "Title")).
>
> * **A4-A5.** Yes. [17] and [18] are duplicated references. Thanks for spotting this and we will correct this properly and streamline the reference format.
>
> * **A6.** To our best knowledge, we have referred necessary publications from 2021 (e.g., [2],[13],[26],[42].[43]) and also presented a comparison with one of SOTA approaches (ProDA [42]) from 2021.
> Probably, we could not find more relevant publications after 2021 as we proposed a pioneering work in distortion-aware domain adaptation.
> Hope this also appears to be reasonable to the reviewer 'ihqk'.

---

> > ### Comment · Reviewer_ihqK · 2022-08-05
> > **Reply**
> >
> > Thanks for the reply, my issues have been basically settled.  I decide to keep my rating on 6.

---

> > > ### Author Response · Authors · 2022-08-08
> > > **Response to ihqk's Reply**
> > >
> > > We thank the reviewer for the positive support and for the time reviewing our work.

---

### Official Review · Reviewer_83ri · 2022-07-11

**Rating:** 7
**Confidence:** 3
**Soundness:** 4 excellent
**Presentation:** 3 good
**Contribution:** 3 good

**Summary:**

This paper proposes a novel segmentation framework that could be aware of distortion caused by optical and geometric reasons and thus improve the segmentation performance using unsupervised domain adaptation. The main contributions of this paper are (1) the framework that can be aware of distortion and utilize the information in the unsupervised semantic segmentation by using distortion-aware domain adaptation, (2) loss functions that effectively enforce the unsupervised domain adaptation and segmentation, and (3) new unsupervised domain adaptation benchmark including pre-existed datasets and newly proposed dataset, where the source and target images have additional domain gaps in the optical distortion.

The distortion-aware domain adaptation is realized by a novel Relative Distortion Learning (R.D.L.) using adversarial training (GAN) and Diffeomprohic transformation. The generator (G) generates deformation fields (visualization of distortion) between the source and target domain images.

The proposed method was evaluated using four datasets, especially two for the source domain, and the other two are utilized as the target domain. The images in the source domain are real-world rectilinear images, and the images in the target domains are optically and geometrically distorted images captured by fisheye images. Here, the authors propose a new dataset called the FDD captured by fisheye cameras (200-degree F.O.V.).

The experiments illustrate the outstanding performance of the proposed distortion-aware domain adaptation framework compared to other state-of-the-art deep learning models and other methods.


**Questions:**

 * Questions & Discussions
1. The definition of the domain adaptation should be more discussed. In this paper, the authors conducted the experiments with the environment where the rectilinear images are the source domain, whereas the images with extreme distortion are the target domain. However, in terms of “general” domain adaptation, the source and the target could be changed. Additionally, both the source domain and the target domain could contain the distortion on the edge. The reviewer is curious that the proposed framework is effective only in a limited environment. Alternatively, the reviewer is curious that other environments should not be discussed in the real-world image. If this case, the authors should clearly discuss this limitation. It is also discussed below [Limitations]-[Major issues].

2. Effectiveness of the discriminator. The reviewer is curious about the effectiveness of the discriminator. Since the generator is well organized in the RDL and the loss function, the adversarial training can cause additional cost rather than a significant improvement in the accuracy. Otherwise, what if the discriminator categorizes the input images into domains (source and target). For instance, what if the discriminator aims to discriminate distortion style between I_T, I_(S->T), I_S, and I_(T->S). Since the generator generates the deformation field (S->T) and the inverse deformation field (T->S), training using four images can lead to the globally optimized discriminator, and it could be more efficient for the framework and the adversarial training. In addition, the ablation study for the discriminator could improve the quality of the manuscript.

3. Cost of the proposed framework. The reviewer is curious about the training time, prediction time, and memory resources. In the manuscript, the authors addressed that the proposed framework is effective compared to the algorithm that removes the distortion from the image. Quantitative analysis or discussion can significantly improve the quality of the manuscript. For the reviewer, since the proposed network includes GAN-based architecture, segmentation-based architecture, and three or more loss functions, the training and prediction cost can be extremely burdened compared to the conventional algorithms.


**Ethics Review Area:**

["I don’t know"]

**Limitations:**

1. Major issues
- The field of interest suggested in this paper is narrow. The reviewers agree that the distortion can significantly degrade the segmentation performance when the image includes extreme distortion like the image captured using a fisheye camera. Even though real-world images contain the distortion, the reviewers could not be convinced that the minor distortion significantly degrades the segmentation performance. It could be justified through more experiments. The authors should significantly address the necessity of distortion-aware domain adaptation in the machine learning society instead of computer vision society. Additionally, the following concepts and experiments should be discussed to justify that the proposed framework is effective in the "more general" domain adaptive segmentation. (1) Utilization of the dataset containing distortion as the source domain and rectilinear images as the target domain. (2) Utilization of the dataset containing distortion as the source domain and another dataset with distortion as the target domain. Note that, this paper discussed the environment where the rectilinear images are utilized as the source domain and the images with distortion as the target domain. Or the authors should justify the experimental environment in this paper.

- In addition, the above questions and the discussion are mainly concerned with a better quality of this paper for Neurips2022 (See Questions).

2. Minor issues
- Motivation for unsupervised learning needs to be more detailed.
As the authors illustrated in Line 45-47 on page 2, the motivation for unsupervised learning is the lack of a real-world public dataset with segmented annotations. Clearly, the lack of annotations for the dataset, including many images, could lead to unsupervised learning. However, that sentence author mentioned in the manuscript implies that the lack of the images (or dataset) leads the unsupervised learning in this paper. As the authors already knew, unsupervised learning with a few images could induce problems with biased training (e.g., overfitting). The reviewer expects that the precise motivation for unsupervised learning should be addressed strongly and clearly and that the authors can improve the quality of the manuscript with few modifications.

- Small batch size should be discussed. As illustrated in Line 30-34, on page 1, in the supplementary file, the batch size for training deep learning models is four until convergence, and the batch size is significantly small. The importance of the normalization methods (e.g., batch normalization or group normalization) depends on the batch size. Therefore, discussing the batch size and the normalization methods would be better.

- The vague description of the dataset and preparations should be improved. (1) the authors mentioned, "Both source and target images are randomly cropped" in the supplementary file in Line 32, on page 1. As the reviewer understood, the images in the target domain include the distortion on edge. At this moment, the random cropping method for the images in the target domain is nonsense. (2) In the experiments, the authors split images into training and validation sets with the random selection method. Is it for the cross-validation or hold-out methods? More explanations and statistical analysis (if authors conducted experiments with many folds) should be discussed even in the supplementary files.

- Justification to search hyper-parameters should be discussed. Common hyper-parameters to train deep learning models are well illustrated in the manuscript and the supplementary files (especially the selection of beta 1-3). However, the reviewer is curious about the selection process of gamma.

- (if possible) Experiments with other evaluation metrics could improve the quality of the manuscript. For instance, the boundary is a significantly important target in the segmentation tasks, and the distortion could degrade the boundary-oriented segmentation accuracy. Therefore, the discussion using evaluation metrics [1-3] for the boundary of the target objects can significantly improve the quality of the manuscript.

[1] Fernandez-Moral, Eduardo, et al. "A new metric for evaluating semantic segmentation: leveraging global and contour accuracy." 2018 IEEE intelligent vehicles symposium (iv). IEEE, 2018.

[2] Lee, Kyungsu, et al. "Boundary-oriented binary building segmentation model with two scheme learning for aerial images." IEEE Transactions on Geoscience and Remote Sensing 60 (2021): 1-17.

[3] Cheng, Bowen, et al. "Boundary IoU: Improving object-centric image segmentation evaluation." Proceedings of the IEEE/CVF Conference on Computer Vision and Pattern Recognition. 2021.

3. Simple recommendations
- Please re-check the typo and grammar errors to improve the quality of the manuscript.
- Complicated quantitative analysis (Tables 5 and 6 in the supplementary material) should be clearly illustrated.
- The reason why the authors illustrated the visualization of the class-activation map in the manuscript should be described in the main manuscript.
- Public access to the code and the dataset should be preceded.

**Strengths And Weaknesses:**

1. Strengths
- The reviewer significantly understands the significance and importance of the task proposed in this paper. The distortion caused by the lens can significantly degrade the segmentation performance, especially the object on edge. In addition, domain adaptive semantic segmentation with lens distortion has rarely been studied to our best knowledge, such that the tasks proposed in this paper for the distortion-aware domain adaptation would be novel. Furthermore, since the dataset construction requires heavy cost, the construction of a large number of datasets for training machine learning models is noteworthy.

- Additionally, the problem definition is clear. The authors defined the domain gap using not only distortion but also visual domain gap. At this point, in terms of the domain gap, the discrepancy between visual reasons and optical and geometric reasons should be discussed, and it should be decomposed. The authors proposed the “only” distortion-aware framework regardless of the visual domain gap.

- The manuscript is well organized and well written.

2. Weakness
- More detailed descriptions are illustrated in the “Limitation” section. Please see below.

---

> ### Author Response · Authors · 2022-08-02
> **Answers to '83ri' (3/3)**
>
> **Answers to other questions**
>
> * **Relatively Small Batch Size:**
> We do not use any batch-normalization layers, following the based methods (AdaptSeg [35] and AdvEnt [36]), since we use a small batch size due to joint training of the deformation field generator and the discriminators along with the segmentation adaptation model.
> In particular, we used the batch size of four so that the training of the models fits an NVIDIA A100 GPU while considering experimental consistency across adaptation tasks for fair comparisons.
>
> * **Pre-Processing and Dataset Split:**
> We are not sure whether we completely understand what *”nonsense”* meant by the reviewer '83ri'.
> As indicated in Line 32 of the supplementary material and above comment on the extension of the tasks ([link](https://openreview.net/forum?id=6RoAxmwj0L2&noteId=IFIbszCvCJN "Title")), we randomly cropped and resized both source and target images.
> In the meanwhile, RDL is able to model relative distortion between any source and target images including randomly cropped and resized images (see Fig.2 and Fig.4 in the manuscript).
> Our extensive experiments also validate the effectiveness of RDL in improving segmentation adaptation using such source and target images.
> Moreover, such data processing is known to be effective for improved semantic segmentation performance [6] and also we followed the pre-processing of the baseline methods for fair comparisons.
> Lastly, we implemented a simple hold-out method where we randomly split the target datasets into the training and the validation set.
>
> * **Hyperparameter $\gamma$:**
> This is a pre-defined hyperparameter to select the baseline adaptation methods where $\gamma=1.0$ implements AdvEnt [36] and $\gamma=0.0$ performs AdaptSeg [35] (see Line 208 in the manuscript).
> So we did not perform an ablation study on $\gamma$.
>
> * **(if possible) Additional Evaluation Metrics:**
> Thanks for suggesting other evaluation metrics that seem to evaluate the boundary-oriented segmentation accuracy.
> Considering the short period of time given for rebuttal, we were not able to implement the metrics and would like to consider them in future work.
> However, please note that the class-wise activation visualizations (Fig.1-(a) in the manuscript and Fig.4 in the supplementary material) show the competence of our method in generating stronger and finer boundary segmentation of objects under severe radial distortion.
> We also demonstrated the competence of our adaptation method in predicting distorted regions by using the distortion-aware mIoU (Fig.5 in the manuscript and Tab.4 in the supplementary material).
>
> * **Public Access to Dataset and Code:**
> We are internally processing the release of the FDD dataset and will release the code upon acceptance.
>
> * We will also re-check typos and grammar errors to improve the quality of the final manuscript.

---

> > ### Comment · Reviewer_83ri · 2022-08-09
> > **Response to Rebuttal**
> >
> > * I would like to express my appreciation to the authors to take the time to my questions.  most of my concerns were resolved in the rebuttal phase; mainly related to the discriminator, the cost of the proposed network, the description of the dataset, and network design (related to batch size). Therefore, I increase my rating from six to seven through the discussion phase. In addition,  I agree that the authors have not constrained the type or direction of the domain adaptation. Despite theoretically unconstrained conditions, however, I would like the authors to exhibit improved performance (in terms of accuracy) through the experiments. The detailed explanations in the Discussion section or a small section could exhibit the possibility of extending the novel methods, the authors proposed, to the general domain adaptation methodology.
> >
> > * Furthermore, for reproducibility and the improvement of the deep learning society, I would strongly address that the code would be published in public. In hopeful expectation, I will adjust my rating in good faith.

---

> > > ### Author Response · Authors · 2022-08-09
> > > **Reply to '83ri's Response to Rebuttal**
> > >
> > > We appreciate your favorable reviews. According to the reviewer's suggestions, we will add a detailed explanation in the Discussion and Conclusion (currently Conclusion section in the manuscript) section discussing the possibility of extending our methods, to the general domain adaptation methodology.
> > > We will definitely consider releasing the code for reproducibility. We are under an internal review process to complete this.

---

> ### Author Response · Authors · 2022-08-02
> **Answers to '83ri' (2/3)**
>
> **Effectiveness of Discriminator ($D_G$)** \
> In training the deformation field generator ($G$), we observed that the output deformation fields $\Phi_{S \rightarrow T}$ and $\Phi_{T \rightarrow S}$ get easily converged to a trivial solution (i.e., identity deformation field $\Phi_{I}$) without introducing the distortion-aware discriminator ($D_G$).
> Such a trivial solution satisfies Eq.(2) and Eq.(3) and ultimately limits the role of the relative deformation field generator $G$ as an identity field generator.
> Then diffeomorphic transformation only reproduces the identical images (e.g., $I_S \cdot \Phi_{S \rightarrow T} = I_S$ when $\Phi_{S \rightarrow T}=\Phi_{I}$).
> To prevent such undesirable local minima, we introduced the discriminator $D_G$ which discriminates the distortion style between $I_T$ and $I_{S \rightarrow T}$.
> We found that the adversarial loss using $D_G$ is effective in learning relative distortion as shown in the ablation results from Tab.3 (${L}_{adv_G}$) in the manuscript.
>
> Please note that we designed a discriminator which primarily targets to minimize the distortion gap between $I_T$ and $I_{S \rightarrow T}$ for the proposed adaptation tasks (from rectilinear source to fisheye target).
> For this, we also tested whether discriminating the distortion style between $I_S$ and $I_{T \rightarrow S}$ helps to improve the segmentation adaptation performance in our early design experiments.
> However, we did not observe further improvements hence we decided to use a single discriminator for $I_T$ and $I_{S \rightarrow T}$.
>
> **Cost of the proposed framework** \
> The optimization goal of our framework is to find the optimal segmentation adaptation model $M^*$ (see Algorithm 1 in the supplementary material).
> The optimal model $M^*$ does not require any additional computational modules at test time and thus the inference time and memory usage remain the same compared to the baseline methods (69.385 ms on a 1280 $\times$ 966 image using about 805.15 MB of an NVIDIA A100 GPU).
> Regarding the training computational costs, as the reviewer '83ri' pointed out, our framework involves additional training of the deformation field generator ($G$), the distortion-aware discriminator ($D_G$), and its adversarial learning (${L}_{adv\_G}$) as well as the distortion-aware losses.
> For example, the based method (AdaptSeg) takes about 9.6 GPU hours of training with 17.98 GB GPU memory, while AdaptSeg+RDL takes about 16.5 GPU hours of training with 20.25 GB GPU memory.

---

> ### Author Response · Authors · 2022-08-02
> **Answers to '83ri' (1/3)**
>
> Above all, we thank the reviewer `83ri' for providing thoughtful and constructive suggestions.
> We hope our below answers address most of your concerns.
>
> **Further clarification of motivation** \
> The reviewer ‘83ri’ expressed curiosity about the various set-ups of domain adaptation tasks among distorted and rectilinear images (e.g., distorted images as source and rectilinear images as a target, or both source and target domains include distorted images).
> Although various adaptation tasks could have been considered, we first tackle adapting existing semantic segmentation models trained on rectilinear images to *unlabeled* fisheye images.
> One of the motivations for our work is the scarcity and the difficulty of constructing *annotations* for distorted fisheye images (e.g., Woodscape [40]), while we already have larger amounts of *annotated* rectilinear images (e.g., Cityscapes [8], GTAV [31]).
> Thus, we first define the scope of our tasks in line with the necessity in real-world scenarios.
> In addition, as the reviewers pointed out, we are not aware of prior works on geometric distortion in domain adaptation.
> As pioneering work in this direction, we believe that we have brought already interesting and practically valuable domain adaptation tasks with extensive experimental results.
> The reviewer 'QLQF' also commented that we introduced new benchmarks that "*could help develop more sophisticated methods that deal with both geometric distortion and appearance change*".
> Similarly, the reviewer '83ri' already suggested an interesting direction where distortion-aware adaptation can be extended to various directions in the machine learning society.
> We are happy to see such constructive suggestions that could have been possible as we brought a new perspective (i.e., geometric distortion shifts) to the domain adaptation field.
>
> **Application to extended domain adaptation tasks** \
> We found the aforementioned extensions of the proposed adaptation tasks plausible to carry out.
> Basically, in our relative distortion learning (RDL), the deformation field generator ($G$, see Fig.1 in the manuscript) produces not only the forward deformation field ($\Phi_{S \rightarrow T}$), but also its inverse ($\Phi_{T \rightarrow S}$); and both fields are utilized in the distortion-aware loss functions (Eq.(2) and Eq.(3)).
> Hence RDL is able to rectify distorted images ($I_{S \rightarrow T}$) to normal images ($I_S'$) via the inverse field ($\Phi_{T \rightarrow S}$), E.g., transform Fig.2-(e) to Fig.2-(d) in the manuscript.
> Also, we do not impose any assumption on the distortion style of source and target images.
> In particular, both target and source images are randomly cropped and resized (see Fig.2 in the manuscript) in our experiments.
> In the meanwhile, RDL is able to address various relative distortions between input images (e.g., both source and target images can have some degree of distortions or not).
> Technically speaking, our distortion-aware adaptation method can be applied to other experimental environments as the reviewer '83ri' suggested and we find this would be one of interesting directions for future work.
>
> We will further clarify the motivation for defining the scope of our work in Introduction; and discuss the possible extensions of adaptation tasks in Conclusion.

---

### Official Review · Reviewer_QLQF · 2022-07-11

**Rating:** 5
**Confidence:** 5
**Soundness:** 2 fair
**Presentation:** 3 good
**Contribution:** 2 fair

**Summary:**

This paper concerns the geometric distortion that causes domain gaps during unsupervised domain adaptation for semantic segmentation, which is motivated by the differences between rectilinear images and fisheye images. It is practically well-grounded.
The authors propose to train a deformation generator that consumes two images chosen from the source and target domains.
The output of the deformation generator is used to map the source image to the target style in terms of geometric distortion.
The training or adaptation is then happened by learning from the distorted source image and semantic segmentations.
The authors also propose a few benchmarks to evaluate the performance of the method and show good domain adaptation gain compared to baselines that concern global appearance change of the images.
An ablation study on each term in the proposed training loss is also performed.

**Questions:**

The main concerns are listed in the above section, please provide more information since they are related to the significance of the proposed problem and method.

some comments and questions on writting
- the title "DaDA: Distortion-aware Domain Adaptation for Unsupervised Semantic Segmentation" may not be appropriate as semantic segmentation is not unsupervised.
- ln 58 "to enforce the semantic quality of relative deformation fields at the image- and the prediction level." how do you define semantic quality of deformation fields? also, what do you mean by "image- and prediction level"?
- ln 142, the equation is confusing, phi_s2t is mapping from the source domain to the target domain, which takes pixel coordinates from source and maps it to source.

**Limitations:**

Did not observe much negative societal impact. On the limitation side, the authors indicate that other methods that deal with texture differences between domains can be combined with the proposed one to further improve performance. However, this is also an indicator that the paper with a clean motivation is evaluating on datasets with multiple causal factors, which does not convey a clean message.

**Strengths And Weaknesses:**

Strength:
- the motivation is practically meaningful and also the introduced setting is interesting.
- the proposed method is simple yet shows good domain adaptation gain. Also, the proposed benchmarks could help develop more sophisticated methods that deal with both geometric distortion and appearance change.
- the constraints to impose semantic consistency of the generated geometric distortion are interesting.
- the paper is well-written and the idea is clearly conveyed.

Weakness:
- even though the motivation is well-grounded, the study of the validity of the problem, or how the motivating factor affects the adaptation performance is not crystal clear. For example, the authors should isolate geometric distortion from other factors like appearance and output space discrepancies. A simple experiment to do is to create a distorted target dataset from rectilinear images, and check how the other methods and the proposed method perform. The current message in this paper does not support a judgment on this aspect.
- In theory, an ideal method should close the gap given only geometric distortion. But there is concern on whether the proposed method con do it under only geometric distortion or not, since the proposed randomly picks two images even though they are not in correspondence. This may be okay, since we can think of the proposed as a kind of spatial augmentation method, but we need to check the results.
- The above concerns are critical for us to analyze the proposed method and the underlying problems that the proposed is trying to solve. For example, affine aug already works well on some of the directions in Table 1, does this mean that spatial augmentation/transformation is the key here? If so, would the proposed method a more sophisticated such augmentation method?
- Also, how does the proposed method compare to spatial transformation generated by optical networks? The proposed do not really care about finding the correct underlying dense matching, which makes OF look like a good candidate to try, say, use OF to output some flow and then warp the images to the target domain or vice versa and then adapt? What is the difference then?
- If camera distortion is the only factor that causes domain gap and is the focus of the current paper, then it seems that camera calibration would be a nice tool and can be performed quite well using existing tools. Then what does the paper tell us beyond that?

---

> ### Author Response · Authors · 2022-08-02
> **Answers to 'QLQF' (2/2)**
>
> **Relationship to Optical Flow** \
> Optical flow (OF) approaches (e.g.,[R2-R4]) commonly require *paired* input images sharing similar contents (e.g., temporally aligned images) to generate a flow field defining the displacement of pixels.
> However, we proposed a relative distortion generator ($G$), which takes a set of *unpaired* source- and target-domain images, to transform the source image into a new image sharing a similar distortion style to the target image.
> We are not aware of any existing OF networks that can be directly employed to produce a flow field that defines relative distortion between unpaired input images without introducing fundamental modifications or inventing ideas as we have proposed.
>
> **Relationship to Camera Calibration** \
> Camera calibration appears to be a simple remedy to address geometric distortion, but it has fundamental disadvantages, including reduced field-of-view (over 30% of pixel losses), resampling distortion artifacts at the periphery, and cumulative calibration errors in practice (see Line 40-44 in the manuscript).
> These are against the purpose of using larger field-of-view cameras and urge us to use native fisheye images instead of considering naive calibration.
> Such disadvantageous in camera calibration has been the core motivation for our work and led to interesting and important domain shift problems as all reviewers commented.
> In addition, our domain adaptation tasks involve not only geometric deformation (e.g., radial distortion) but also visual domain shifts (e.g., texture, lighting, contrast) between source and target images.
> We are also delighted to have a comment from the reviewer `QLQF' saying that such problems are *“practically meaningful”* and *“could help develop more sophisticated methods”* in the follow-up research.
>
> **Other Comments and Questions**
>
> * **Title of the paper**\
> Basically, we try to solve an unsupervised domain adaptation problem for semantic segmentation, where we assume the target images do not have corresponding annotations. Thus, we used *”unsupervised”* term in the title.
> However, if the reviewer still has a concern about this term, we will consider moving *”unsupervised”* to before *”Domain Adaptation”* which makes the title "DaDA: Distortion-aware Unsupervised Domain Adaptation for Semantic Segmentation".
>
> * **Clarifications of terminologies**
>   * **Line 58**: we evaluate *“semantic quality”* of deformation fields by how much they can reduce the distribution shifts across domains since it is very challenging to quantitatively measure.
> Also, we are not aware of any existing methods directly applicable for distortion similarity measures under *unpaired* sets of images.
> Instead we try to enforce the semantic quality at both *”image level”* by aligning the distortion style of transformed source images $I_{S \rightarrow T}$ with $I_T$ in Eq.(4) and Eq.(5) and *”prediction level”* by minimizing consistency between the predictions, $M(I_T)$, $M(I_{S \rightarrow T})$, $M(I_S)$, $M(I_{T \rightarrow S})$, in Eq.(3).
>   * **Line 142**: $\Phi_{S \rightarrow T}$ maps pixel coordinates of a source image to those of a new image $I_{S \rightarrow T}$ which shares a similar distortion style of $I_T$ (see Line 139-141).
>
> **References** \
> [R2] Xu et al., "GMFlow: Learning Optical Flow via Global Matching." Proceedings of the IEEE/CVF Conference on Computer Vision and Pattern Recognition. 2022. \
> [R3] Luo et al., "Upflow: Upsampling pyramid for unsupervised optical flow learning." Proceedings of the IEEE/CVF Conference on Computer Vision and Pattern Recognition. 2021. \
> [R4] Wang et al., "Displacement-invariant matching cost learning for accurate optical flow estimation." Advances in Neural Information Processing Systems. 2020.

---

> > ### Comment · Reviewer_QLQF · 2022-08-09
> > **only partially resolved**
> >
> > Thanks for the authors' response. However, important questions are still not resolved. Please check the following.
> > To be explicit, I am not worrying that the proposed can not do better than other adaptation methods that mainly concern appearance gaps.
> > However, my concern lies in two folds.
> > First, how significant is the role of the learnable spatial distortion network compared with spatial distortion augmentation that does not need to train networks? Because in any case, you do not care about exact correspondence, and for OF we have SIFT-flow or semantic flow, which does not require paired images.
> > Second, I am not convinced that doing camera calibration will be such a disadvantage, e.g., occlusions can be resolved using an indicator mask. Is there any data showing that calibration would be introducing errors even when handled correctly?
> > Given the current information, I like to keep my rating.

---

> > > ### Author Response · Authors · 2022-08-09
> > > **Reply to 'QLQF's Response to Rebuttal**
> > >
> > > We thank the reviewer ‘QLQF’ for the informative replies. We hope our following answers address your concerns properly.
> > >
> > > Regarding suggested optical flow (OF) methods, we found Semantic-Flow [R5] very interesting as it exploits localized layer information from semantic segmentation to further improve optical flow tasks. We also noted that SIFT-Flow [R6] considers *unpaired* scenes for motion synthesis via object transfer.
> > > However, prior works on optical flow (e.g., [R5],[R6]) primarily aim to solve *motion* prediction between images. Even though some of optical flow methods allow *unpaired* sets of images, it remains unknown how the non-learnable image descriptor (e.g., SIFT) for *explicit* local correspondence matching can be directly applied to *implicit* relative distortion learning for distortion style transfer.
> > > Moreover, we cannot say that our method ignores the correspondence. For example, Fig.2 and Fig.4 in the manuscript show that buildings and vehicles are distorted by replicating counterparts in target images by implicit learning of distortion correspondence (RDL) (Line 301-308 in the manuscript).
> > > Lastly, throughout our experiments, we have shown the importance of our relative distortion learning (RDL) for distortion-aware segmentation adaptation, where +RDL (*learnable*) always improves adaptation performance while +RA (randomized affine augmentation) leads to degraded performance in some cases (e.g., +RDL vs. +RDL+RA in *Cityscapes $\rightarrow$ CityscapesFisheye* and *Cityscapes $\rightarrow$ FDD*).
> > > Above all, we thank the reviewer 'QLQF' for constructive suggestions, and we would like to investigate the possibilities of optical flow as spatial augmentation methods in the follow-up research.
> > >
> > >
> > > Regarding the second question about camera calibration, in Line 40-43 of the manuscript, we referred to Kumar et al. [20], where they experimentally evaluated the disadvantage of calibrations including reduced field-of-view, resampling error, and calibration errors in practice (*"Practical Problems encountered"* section in [20]). Hope this appears to be reasonable to the reviewer 'QLQF'.
> > >
> > >
> > >
> > > References:
> > >
> > > [R5] Sevilla-Lara et al. "Optical flow with semantic segmentation and localized layers." Proceedings of the IEEE conference on computer vision and pattern recognition. 2016.
> > >
> > > [R6] Liu et al. "Sift flow: Dense correspondence across scenes and its applications." IEEE transactions on pattern analysis and machine intelligence 2010
> > >
> > > [20] Kumar et al. "Unrectdepthnet: Self-supervised monocular depth estimation using a generic framework for handling common camera distortion models." 2020 IEEE/RSJ International Conference on Intelligent Robots and Systems. IEEE, 2020.

---

> ### Author Response · Authors · 2022-08-02
> **Answers to 'QLQF' (1/2)**
>
> We thank the reviewer `QLQF' for a thorough and constructive review.
> Below, we have answered your questions and concerns regarding the effect of the geometric distortion and other aspects of the submission.
> Hope they appear to be reasonable to you too.
>
> **Effect of Geometric Distortion on Unsupervised Domain Adaptation (UDA)** \
> To clarify the effect of geometric distortion on the adaptation tasks, the reviewer ‘QLQF’ suggested performing an experiment where the geometric distortion is isolated from other factors in distributional shifts (e.g., visual domain gaps).
> For example, ‘QLQF’ presented an experiment where we synthetically generate distorted target images from rectilinear source images, and then evaluate the proposed method with other approaches.
>
> Actually, in our early stage of development, we executed a similar adaptation experiment to preliminary validate our distortion adaptation approach under only geometric distortion in domain shift.
> There we took the Cityscapes dataset as source and its distorted counterpart as target (*CityscapesFishEye* includes fisheye-like images similar to $I_T$ in Fig.1 in the manuscript).
> The distorted images are generated based on the equidistance fisheye camera projection model [R1].
> To address 'QLQF's concerns, we performed the *Cityscapes $\rightarrow$ CityscapesFishEye* adaptation task again with the exact same experimental set-up for *Cityscapes $\rightarrow$ Woodscape* task.
> We present the results in the following table.
> \
> \
> **Results from the *Cityscapes $\rightarrow$ CityscapesFisheye* adaptation task**
> | **Method**                      | **mIoU(%)**   | **gain**      |
> |---------------------------------|----------------|---------------|
> | Oracle (trained on target)      | 69.82          | -             |
> | Source Only (trained on source) | 35.69          | -             |
> ||||
> | AdaptSeg [35]                   | 47.16          | -         |
> | AdaptSeg+**RDL**                    | 57.89      | +10.73     |
> ||||
> | AdaptSeg+RA                     | 54.02          | -         |
> | AdaptSeg+RA+**RDL**                 | 55.58          | +1.56         |
> ||||
> | AdvEnt [36]                     | 46.67          | -         |
> | AdvEnt+**RDL**                      | 57.04      | +10.37     |
> ||||
> | AdvEnt+RA                       | 54.55          | -         |
> | AdvEnt+RA+**RDL**                   | 55.82          | +1.27         |
>
> Results clearly show that our relative distortion learning (RDL) contributes to significant improvements in the adaptation performance up to +10.73% when only geometric distortion is presented in the distributional shifts.
> This is somewhat obvious to observe since the baseline methods (i.e., AdaptSeg [35], AdvEnt [36]) do not consider the geometric distortion in domain shifts while our approach features distortion-aware adaptation based on relative distortion learning (RDL).
>
> Remarkably, +RDL achieves the largest gain over the based method and such results are echoed in the *Cityscapes $\rightarrow$ FDD* task in Tab.1 of the manuscript.
> Please note that RDL always leads to improvements in segmentation adaptation, regardless of domain shift, throughout our experiments (e.g., based methods vs. +RDL, and +RA vs. +RA+RDL) as presented in Tab.1 of the manuscript and above result table.
> In contrast, the randomized affine augmentation (RA) leads to degraded segmentation adaptation results upon the geometric distributional shifts between source and target domains (e.g., +RDL vs. +RDL+RA in *Cityscapes $\rightarrow$ CityscapesFisheye* and *Cityscapes $\rightarrow$ FDD*).
> Thus, we may state that our *learnable diffeomorphic transformation* (RDL) plays an important role in aligning the domain gap of geometric deformation.
>
> We will add the results from *Cityscapes $\rightarrow$ CityscapesFisheye* to Additional Experimental Results (Sec.B) in the supplementary material and clarify the aforementioned discussion on the effect of +RDL in Comparisons with State-of-the-Art Methods (Sec.4) in the manuscript.
>
> **Reference** \
> [R1] Kannala et al., “A generic camera model and calibration method for conventional, wide-angle, and fish-eye lenses” IEEE transactions on pattern analysis and machine intelligence 2006.

---

### Meta-Review · Area_Chair_H7KF · 2022-08-23

**Recommendation:** Accept
**Confidence:** Certain

**Metareview:**

This paper proposes a new segmentation method with geometric insight to deal with the distortions.  As pointed out by our reviewers, this paper is featured with important practical value, clear problem definition, and interesting mathematical insight. During the rebuttal phase, most of the reviewers confirmed their support for an (weak) acceptance, and I believe this paper should be accepted as a poster paper.

**Award:**

No

---

### Decision · Program_Chairs · 2022-09-14

Accept